# PRL3-zumab as an immunotherapy to inhibit tumors expressing PRL3 oncoprotein

Min Thura[1,12], Abdul Qader Al-Aidaroos [1,12], Abhishek Gupta[1], Cheng Ean Chee[2], Soo Chin Lee[2], Kam Man Hui[3], Jie Li[1], Yeoh Khay Guan[4], Wei Peng Yong[2], Jimmy So[5], Wee Joo Chng[2], Chin Hin Ng[2], Jianbiao Zhou [2], Ling Zhi Wang[6], John Shyi Peng Yuen[7], Henry Sun Sien Ho[7], Sim Mei Yi [7], Edmund Chiong[5], Su Pin Choo[8], Joanne Ngeow[1,8,9], Matthew Chau Hsien Ng[8], Clarinda Chua[8], Eugene Shen Ann Yeo[10], Iain Bee Huat Tan[8], Joel Xuan En Sng[1], Nicholas Yan Zhi Tan[1], Jean Paul Thiery[1], Boon Cher Goh[2] & Qi Zeng[1,11]

Tumor-specific antibody drugs can serve as cancer therapy with minimal side effects. A humanized antibody, PRL3-zumab, specifically binds to an intracellular oncogenic phosphatase PRL3, which is frequently expressed in several cancers. Here we show that PRL3-zumab specifically inhibits PRL3+ cancer cells in vivo, but not in vitro. PRL3 antigens are detected on the cell surface and outer exosomal membranes, implying an 'inside-out' externalization of PRL3. PRL3-zumab binds to surface PRL3 in a manner consistent with that in classical antibody-dependent cell-mediated cytotoxicity or antibody-dependent cellular phagocytosis tumor elimination pathways, as PRL3-zumab requires an intact Fc region and host FcγII/III receptor engagement to recruit B cells, NK cells and macrophages to PRL3+ tumor microenvironments. PRL3 is overexpressed in 80.6% of 151 fresh-frozen tumor samples across 11 common cancers examined, but not in patient-matched normal tissues, thereby implicating PRL3 as a tumor-associated antigen. Targeting externalized PRL3 antigens with PRL3-zumab may represent a feasible approach for anti-tumor immunotherapy.

[1] Institute of Molecular and Cell Biology, Agency for Science, Technology and Research (A*STAR), Singapore 138673, Singapore. [2] Department of Haematology-Oncology, National University Cancer Institute, Singapore (NCIS), Singapore 119082, Singapore. [3] Division of Cellular and Molecular Research, National Cancer Centre Singapore, Singapore 169610, Singapore. [4] Department of Medicine, Yong Loo Lin School of Medicine, National University of Singapore, Singapore 119260, Singapore. [5] Division of Surgical Oncology, National University Cancer Institute, Singapore (NCIS), Singapore 119082, Singapore. [6] Cancer Science Institute of Singapore, National University of Singapore, Singapore 117599, Singapore. [7] Department of Urology, Singapore General Hospital, Singapore 169608, Singapore. [8] Division of Medical Oncology, National Cancer Centre Singapore, Singapore 169610, Singapore. [9] Lee Kong Chian School of Medicine, Nanyang Technological University, Singapore 308232, Singapore. [10] Department of Colorectal Surgery, Singapore General Hospital, Singapore 169608, Singapore. [11] Department of Biochemistry, Yong Loo Lin School of Medicine, National University of Singapore, Singapore 119260, Singapore. [12] These authors contributed equally: Min Thura, Abdul Qader Al-Aidaroos. Correspondence and requests for materials should be addressed to Q.Z. (email: mcbzengq@imcb.a-star.edu.sg)

A major challenge in cancer therapy is the lack of drug specificity. Many cancer-associated targets of currently approved drugs are also often expressed in normal tissues, inadvertently causing off-target tissue damage. Consequentially, the "holy grail" of cancer research has been the exploration of more efficacious therapies against druggable, tumor-specific oncotargets[1]. Antibody-based therapies have proven superior to standard chemotherapy in precisely targeting malignant cells with reduced side effects, acting like magic bullets[2]. Therefore, for a breakthrough in tumor-specific cancer therapy, safety and efficacy, there is an urgent need to continuously identify additional tumor-specific antigens targetable by precise antibody-based drugs.

Phosphatase of regenerating liver 3 (PRL3 or PRL-3) belongs to a unique family of C-terminal prenylated phosphatases within the protein tyrosine phosphatase superfamily, consisting of 3 closely-related members—PRL1, PRL2, and PRL3[3]. In 2001, the Vogelstein group showed that PRL3 was overexpressed in metastatic liver lesions compared to corresponding primary colorectal tumors or normal colon epithelium[4]. PRL3 upregulation has subsequently been reviewed to show ubiquitous correlation with advanced cancers and poorer prognosis[5]. Traditionally, to target intracellular oncoproteins such as PRL3, small chemical inhibitors (rather than antibodies) are screened in in vitro systems as the first-line assay for anti-cancer cell activity, primarily because intracellular compartments are presumed to be inaccessible to large antibody molecules. Since antibodies are more specific and cause fewer side effects than small chemical compounds[1], our group has actively worked for more than a decade at the forefront of unconventional immunotherapy using antibodies (rather than small chemical inhibitors) to block tumors expressing PRL3, as well as other intracellular oncoproteins, in various animal models[6]. Challenging the dogma, we first proved that intravenously administered PRL3 or PRL1 antibodies could block metastatic lung tumors expressing intracellular PRL3 or PRL1 oncoproteins[7]. In follow-up studies, we established a general concept of targeting multiple intracellular oncoproteins with antibodies from different species or vaccination in several animal models[8–11]. Today, this concept has been recognized as an emerging field of cancer immunotherapy[12,13]. In 2016, we proved that a humanized PRL3 antibody (PRL3-zumab; IgG1) could suppress PRL3+ gastric tumors in a clinically relevant orthotopic model for evaluating drug efficacy[11,14]. Collectively, these reproducible findings cement the targetability of intracellular oncoproteins using antibodies[15,16].

Hepatocellular carcinoma (HCC), the most common type of liver cancer, is the second main cause of cancer-related mortalities and fifth most common cancer worldwide[17]. Herein, we explored the utility of PRL3-zumab in treating HCC, a disease with frequent PRL3 overexpression[18,19], and with an unmet need for efficacious and well-tolerated targeted drugs[20].

Using clinically relevant orthotopic liver "seed and soil" tumor models, we show that PRL3-zumab specifically inhibits PRL3+ liver tumors. In freshly dissociated tumor tissues, we demonstrate that PRL3 can be detected on the surface of live tumor cells, a phenomenon that can be partially recapitulated on serum-starved cancer cells in vitro where PRL-3 also localizes to the outer surface of secreted exosomes. The presence of surface PRL3 antigens suggests that PRL3-zumab recognizes and targets PRL3+ tumors for elimination in a similar manner as antibodies against classical extracellular targets, as implicated by the requirement of intact Fc region in PRL3-zumab to interact with host FcγRs for recruitment of immune effectors and effective elimination of PRL3+ tumors. Finally, we showcase the clinical relevance of PRL3 as a frequently expressed tumor antigen across 11 major cancer types globally, warranting the exploration of PRL3-zumab as a potential drug against these common malignancies.

## Results

**In mice, PRL3-zumab blocks PRL3+ liver tumors.** Orthotopic tumor models, wherein human cancer cells ("seeds") are implanted into the organs ("soil") from which the cancer originated, replicate human disease with high fidelity and more accurately recapitulate clinically relevant therapeutic responses[21]. To dissect the mechanism of how PRL3-zumab could target tumors that express intracellular PRL3, in this study, we established an orthotopic liver model to test the ability for PRL3-zumab to block liver tumors within their natural niche. In a panel of six human (Fig. 1a, lanes 1–6) and two murine (Fig. 1a, lanes 7–8) liver cancer cell lines screened for PRL3 protein expression status, we identified three human liver cancer cell lines—MHCC-LM3, Huh-7, and Hep3b2.1 (Fig. 1a, lanes 1, 3, and 5)—which expressed endogenous PRL3 (20 kDa). However, only MHCC-LM3 (Fig. 1a, lane 1) could robustly form sizeable orthotopic liver tumors within our manageable short timeframe (≤5 weeks). The MHCC-LM3 cell line was used as the most suitable model for studying PRL3-zumab therapeutic response in view of two important factors: (1) PRL3 positivity and (2) rapid orthotopic liver tumor formation. Figure 1b illustrates our orthotopic liver model wherein liver cancer cells were implanted into mouse liver to generate orthotopic liver tumors, followed by treatment with PRL3-zumab for 5 weeks (2 doses per week). Compared to untreated mice, MHCC-LM3 liver tumor formation in PRL3-zumab -treated mice was visibly reduced and measurement of tumor volumes revealed a significant, 7-fold reduction in mean tumor burden between treated mice and untreated mice (Fig. 1c). To study if PRL3-zumab could extend mice survival beyond the 5-week treatment duration, independent groups of treated and untreated mice were monitored post treatment until the appearance of morbid characteristics (classified as a death event). Clearly, treated mice had a longer median survival time of 12 weeks compared to 8 weeks for untreated mice (Supplementary Fig. 1). Since both PRL3− HepG2 and SNU449 human liver cancer cell lines (Fig. 1a, lanes 2 and 4, respectively) could not form tumors within 5 weeks, we next utilized the highly tumorigenic Hep53.4 murine PRL3− liver cancer cell line as a negative control for PRL3-zumab therapy (Fig. 1a, lane 7), as it robustly forms sizeable liver tumors within 5 weeks. In addition, we engineered a Hep53.4 cell line overexpressing an enhanced green fluorescence protein-tagged PRL3 fusion protein (EGFP-PRL3; 45 kDa) with forced PRL3 expression (Hep53.4-PRL3; Fig. 1a, lane 8) and established Hep53.4 (PRL3−) or Hep53.4-PRL3 (PRL3+) orthotopic liver tumors in mice for PRL3-zumab treatment. As expected, PRL3-zumab failed to inhibit Hep53.4 tumors that lacked PRL3 expression, with no differences observed between treated and untreated mice (Fig. 1d). In contrast, PRL3-zumab strongly inhibited Hep53.4-PRL3 tumors, with significant differences between treated and untreated mice (Fig. 1e). Collectively, these findings from orthotopic liver models reinforce the fundamental principle that PRL3-zumab therapy specifically blocks PRL3+ (but not PRL3−) tumors.

**In culture, PRL3-zumab does not inhibit PRL3+ cancer cells.** Because PRL3-zumab blocked PRL3+ tumors in vivo, we then investigated if PRL3-zumab could inhibit the growth of PRL3+ cancer cells in vitro: MHCC-LM3 (PRL3+; Fig. 1c) or Hep53.4-PRL3 cells (ectopic PRL3+; Fig. 1e) by adding high doses (up to 50 μg mL−1) of PRL3-zumab directly to cultured cells. Regardless of dose or PRL3 expression status in this simplified in vitro system, we found that PRL3-zumab had no inhibitory effects (Fig. 1f–h, red boxes) on the growth of MHCC-LM3 cells (PRL3+; Fig. 1f), Hep53.4 cells (PRL3−; Fig. 1g), or Hep53.4-PRL3 cells (PRL3+; Fig. 1h). In contrast, cisplatin, a well-known chemotherapeutic

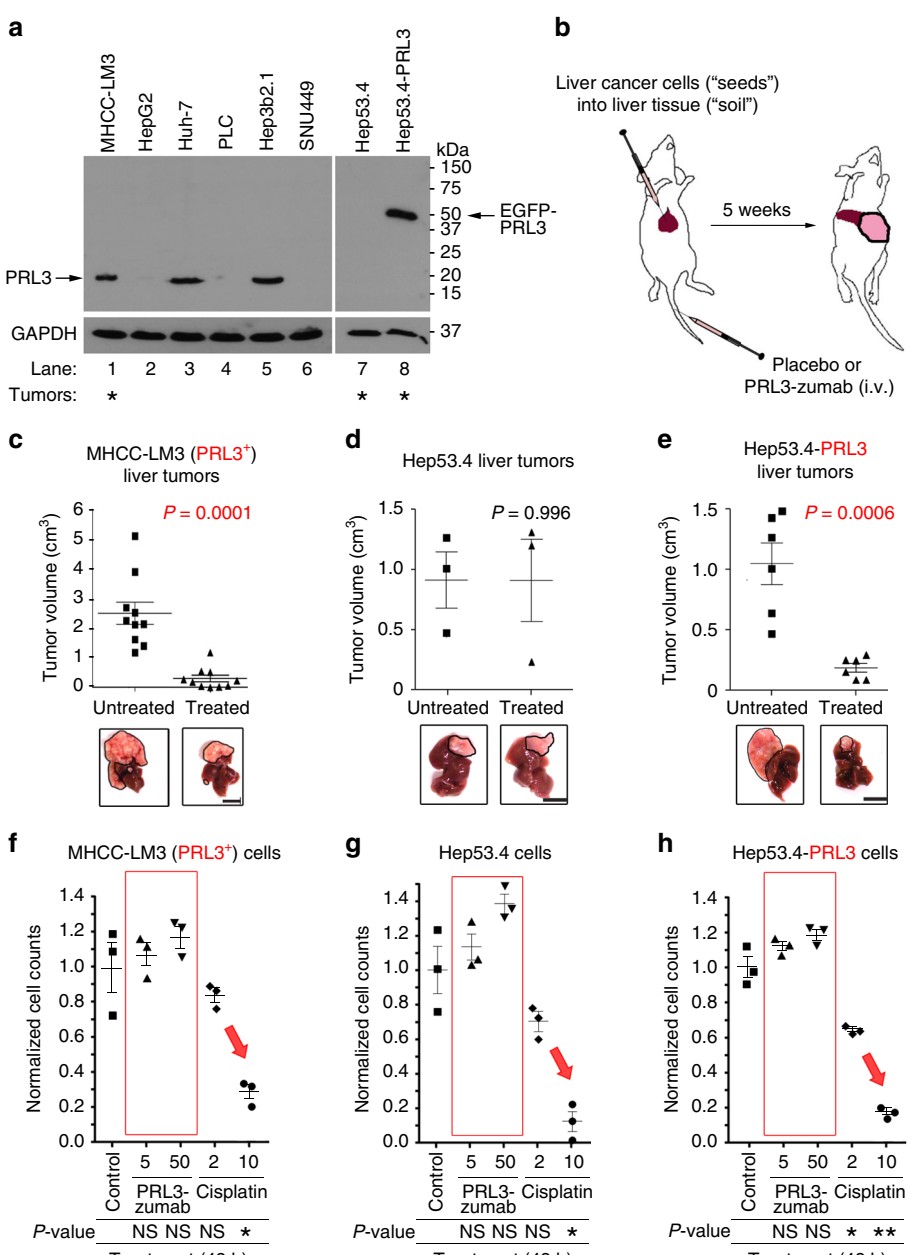

**Fig. 1** PRL3-zumab inhibits PRL3⁺ liver tumors in vivo but not cancer cells in vitro. **a** Representative western blot (WB) of PRL3 protein expression in human (lanes 1–6) and murine (lanes 7 and 8) liver cancer cells. Glyceraldehyde 3-phosphate dehydrogenase (GAPDH) served as a loading control. Asterisks indicate cell lines that rapidly generate orthotopic liver tumors within 5 weeks. **b** Outline of orthotopic "seed and soil" liver tumor model for treatments. **c–e** Mean volumes at the end of the experiment in treated (filled squares) and untreated (filled triangles) groups of mice bearing PRL3⁺ MHCC-LM3 tumors ($n = 10$ mice per group; **c**), PRL3⁻ Hep53.4 tumors ($n = 3$ mice per group; **d**), and Hep53.4-PRL3 tumors ($n = 6$ mice per group; **e**). The mean value was calculated by the Student's $t$ test (mean ± s.e.m.). $P$ values between treatment pairs as indicated. Lower panels, representative liver tumors at the end of experiment. Scale bar, 10 mm. **f–h** The viabilities of MHCC-LM3 cells (**f**), Hep53.4 cells (**g**), and Hep53.4-PRL3 cells (**h**) cultured for 48 h with PBS control (filled squares), 5 µg mL⁻¹ PRL3-zumab (filled upright triangles), 50 µg mL⁻¹ PRL3-zumab (filled inverted triangles), 2 µg mL⁻¹ cisplatin (filled diamonds), or 10 µg mL⁻¹ cisplatin (filled circles) were evaluated by an MTS (3-(4,5-dimethylthiazol-2-yl)−5-(3-carboxymethoxyphenyl)−2-(4-sulfophenyl)-2H-tetrazolium) assay. The mean value was calculated by the Student's $t$ test (mean ± s.e.m., $n = 3$ biologically independent samples each). $^*P < 0.05$, $^{**}P < 0.01$, NS, not significant, as compared between treatment and control group for each cell line. Source data are provided as a Source Data file

drug, resulted in a nonspecific dose-dependent growth inhibition in all three cell lines (Fig. 1f–h, red arrows). These findings indicated that PRL3-zumab only works in vivo, reflecting the requirement of host immune system for its therapeutic effect[6].

**PRL3-zumab binds to externalized surface PRL3 antigens.** As PRL3-zumab could inhibit tumors expressing PRL3 in mice, we investigated if intracellular PRL3 antigen could be externalized

in vivo as an extracellular target for PRL3-zumab binding. Orthotopic PRL3⁺ MHCC-LM3 liver tumors were freshly harvested and dissected into live, single-cell suspensions to compare the percentages of surface PRL3 expression on these ex vivo tumor cells vs. parental MHCC-LM3 cultured cells using PRL3-zumab or cetuximab, a well-known chimeric antibody against the epidermal growth factor receptor (EGFR) as a positive surface protein control (Fig. 2a). Live and dead cells were also

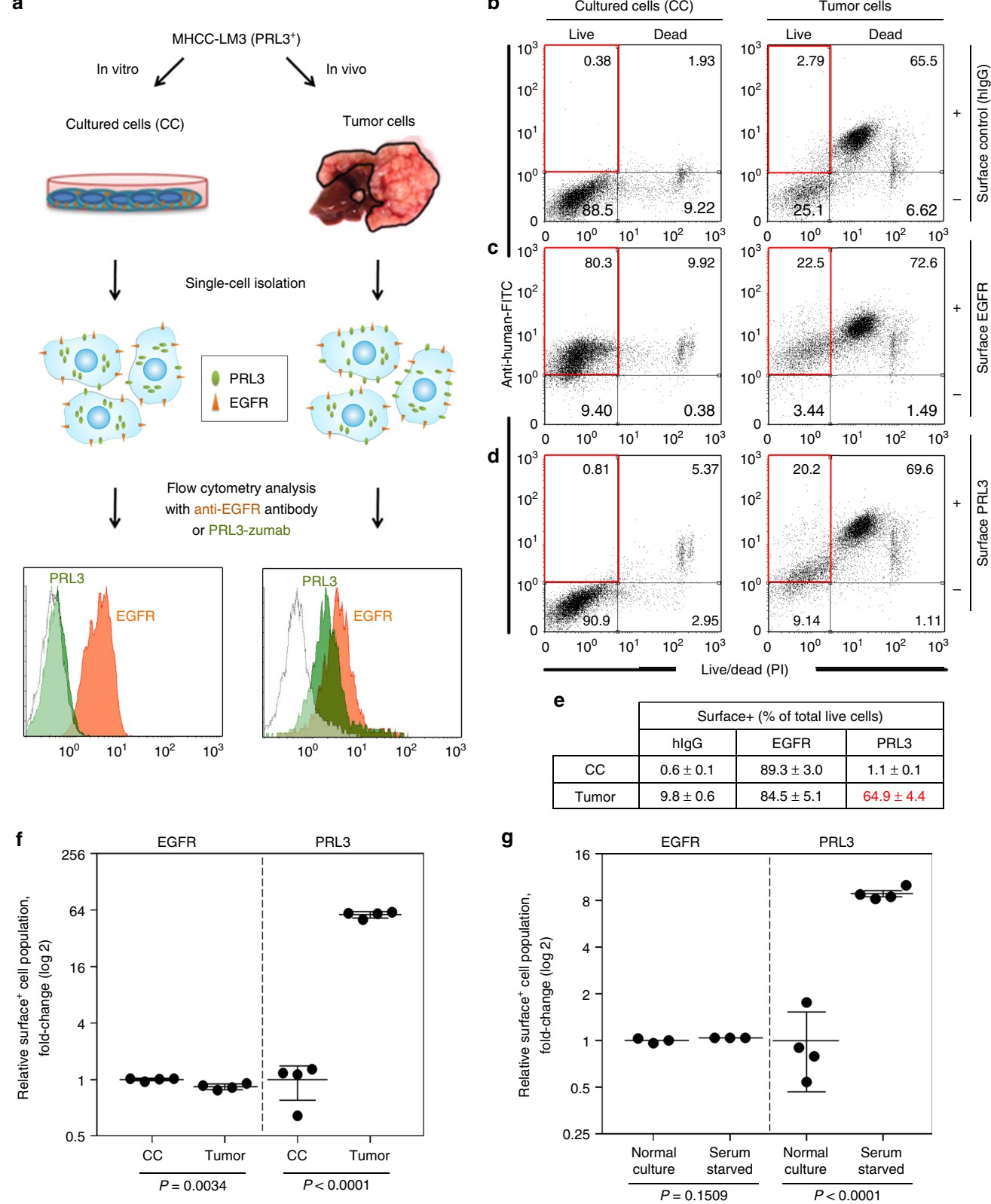

distinguished using the nonspecific uptake of fluorescent dyes, which freely passes through the compromised membranes of non-viable (dead) cells and thus labels them brightly. As dead cells had pronounced nonspecific uptake of control human IgG (hIgG) antibodies (Fig. 2b; top right quadrants), we only considered surface antigen expression on live cells (Fig. 2b–d; top left quadrants) for the rest of this study. The percentage of surface positive (surface+) cells was subsequently calculated based on the number of surface+ live cells (upper left quadrants) divided by the total live cells (sum of upper and lower left quadrants; Fig. 2e).

**Fig. 2** Intracellular PRL3 is externalized for PRL3-zumab binding. **a** Methodology for cell surface analysis of MHCC-LM3 cultured cells (CC) and tumor cells. **b**–**d** "Surface" detection of nonspecific control antigens (**b**), EGFR (**c**), or PRL3 (**d**) were detected by fluorescence-activated cell sorting (FACS) analysis using polyclonal human hIgG, anti-EGFR antibody (cetuximab), or anti-PRL3 antibody (PRL3-zumab), respectively. Representative FACS profiles from four biological replicates are shown. **e** Mean percentage ± s.d. of surface positive (surface+) live cells for each antigen were calculated by dividing the surface antigen-positive live cells (upper left quadrant) by total live cells (sum of both upper and lower left quadrants) in **b**–**d**. **f** Background-corrected values from **e** were normalized to CC surface expression levels for EGFR (filled circles) and PRL3 (filled squares). The mean fold change was calculated by the Student's $t$ test (mean ± s.d., $n = 4$ biologically independent samples). $P$ values as indicated for each antigen. **g** Background-corrected values of MHCC-LM3 cells cultured under "Normal" vs. "Serum-starved" conditions for 72 h were normalized to "Normal" surface+ cell percentages for each antigen. The mean fold-change was calculated by the Student's $t$ test (mean ± s.d.) for EGFR (filled circles; $n = 3$ independent samples) and PRL3 (filled squares; $n = 4$ independent samples). $P$ values as indicated for each antigen. Source data are provided as a Source Data file

Unlike EGFR, which was abundantly expressed on both cultured and tumor cells (Fig. 2c–e), surface PRL3 was poorly expressed on cultured cells, yet expressed on most tumor cells (1.1% vs. 64.9% respectively; Fig. 2d–e). After correcting for nonspecific binding signals, EGFR surface+ cell populations between tumor and cultured cells were comparable (Fig. 2f), while a 57-fold increase in mean PRL3 surface+ cell population was noted in tumors compared to cultured cells (Fig. 2f). Consistent with MHCC liver tumors, we observed a similar increase in PRL3 surface+ cells in metastatic lung tumors formed by tail vein injection of B16F0 melanoma cells (Supplementary Fig. 2a), which express endogenous PRL3 and respond well to PRL3 antibody therapy[8]. The PRL3 surface+ cell population was upregulated 14-fold on tumor isolates from these metastatic PRL3+ B16F0 melanoma lung tumors compared to the cognate cultured cells (Supplementary Fig. 2b). These results show that upregulation of the PRL3 surface+ cell population could be a common feature of PRL3+ solid tumors in vivo. Since total expression levels of PRL3 are only slightly higher in tumor isolates compared to cultured cells (Supplementary Fig. 3a), our findings suggest that the increase in PRL3 surface+ cell populations could be attributed primarily to enhanced PRL3 relocalization rather than an increase in absolute PRL3 expression per se.

Since mechanical and enzymatic tumor dissociation ex vivo might induce cell death or membrane damage (liver tumors, in particular, are considered as "tough" tissues based on their histological composition and require extended treatment time), we next considered whether the increase in PRL3 surface+ cell populations observed might be related to apoptotic induction. Although early apoptotic cells may still have intact cellular membranes and could thus appear "live" in our Live/Dead analysis, they can be readily identified using Annexin-V, which specifically binds phosphatidylserine, a phospholipid extensively "flipped" onto the outer plasma membranes of early apoptotic cells[22]. Using EGFR as a positive surface protein control, we found that 15–25% of both EGFR surface+ and PRL3 surface+ "live" tumor cells were viable (Annexin-V−), whereas the remaining population were in early stages of apoptosis (Annexin-V+; Supplementary Fig. 3b, 3c). These results validate that, like EGFR, surface PRL3 is naturally expressed on viable tumor cells, and its externalization does not depend on apoptosis.

The microenvironment of solid tumors is characterized by numerous stressors, including nutrient deprivation, low pH, hypoxia, and oxidative stress[23]. We hypothesized that the difference in PRL3 surface+ cell populations between cultured and tumor cells might be due to a limitation of standard, empirically defined culture conditions to faithfully recapitulate such stresses present within the tumor microenvironment. To investigate the possible influence of microenvironmental stress conditions on surface PRL3 expression in vitro, we serum-starved MHCC-LM3-cultured cells as a simplified model of an in vivo stress faced by solid tumors and assayed for expression of both EGFR and PRL3 on live cells (Supplementary Fig. 3d, e).

Prolonged serum starvation of MHCC-LM3 cells for 72 h did not induce significant changes in EGFR surface+ cell population (Fig. 2g), whereas PRL3 surface+ cell population increased 8.4-fold upon serum starvation (Fig. 2g). Interestingly, at the molecular level, we detected antagonistic activation of pro-survival vs. pro-apoptosis and autophagy pathways upon serum starvation (Supplementary Fig. 4), resulting in a complex milieu that might enhance PRL3 externalization in starved cells. Likewise, we reasoned that the upregulation of PRL3 surface+ population was greater in tumor cells (57-fold; Fig. 2f) compared to serum-starved cultured cells (8.4-fold; Fig. 2g) likely due to the additional stresses faced within the tumor microenvironment, such as hypoxia or pH stress, which might further exacerbate PRL3 surface relocalization. Taken together, we provide evidence for stress-inducible cell surface relocalization of intracellular PRL3 antigens to demonstrate mechanistic support for PRL3-zumab's ability to recognize and target PRL3+ tumor cells in vivo.

**PRL3 may be externalized via the exosomal secretion pathway.** Since PRL3 lacks a signal sequence that could direct it across the classical endoplasmic reticulum–Golgi secretory pathway, a key question was how PRL3 could be recruited from the cytoplasmic leaflets of the plasma membrane and/or early endosomes to the outer leaflet of the plasma membrane to be localized on the tumor cell surface. Numerous intracellular proteins, including heat-shock protein 70 (HSP70), heat-shock protein 90 (HSP90), and glucose-regulated protein 78 (GRP78), have been reported to be specifically relocalized to the cell surface only in tumor cells, but not in normal cells[24]. In addition, while apoptosis and necrosis could result in leakage and relocalization of intracellular antigens, antibodies against intracellular gp75 can reject tumors where there is no necrosis, suggesting alternative specific pathway(s) enabling antigen externalization for antibody binding[25]. This could include unconventional protein transport pathways, such as exosome secretion, which occurs independently of the endoplasmic reticulum and Golgi to enhance protein translocation to the tumor cell surface[26].

As exosomes have been characterized to stem from endosomes[27], a compartment where PRL3 protein accumulates[3], we investigated if metastasis-associated PRL3 might also be externalized via the exosomal pathway. To avoid contamination from exosomes abundantly present in fetal bovine serum used for cell culture, we assayed for endogenous PRL3 expression in exosomes harvested from MHCC-LM3 (PRL3+) and Hep53.4 (PRL3−) cells cultured under serum-free conditions for 24 h. Endogenous PRL3 readily accumulated in exosomes secreted specifically by MHCC-LM3 cells but not by Hep53.4 cells (Fig. 3a, lanes 1–4). Exosomal enrichment was validated using the focal adhesion protein, paxillin (exosome-negative), and the ESCRT-1 complex protein component, TSG101 (exosome-positive; Fig. 3a). To test if PRL3 might possess specific localization signal(s) for packaging into exosomes, we transiently transfected MHCC-LM3 cells with GFP (control) or GFP-tagged PRL3 to generate MHCC-GFP and

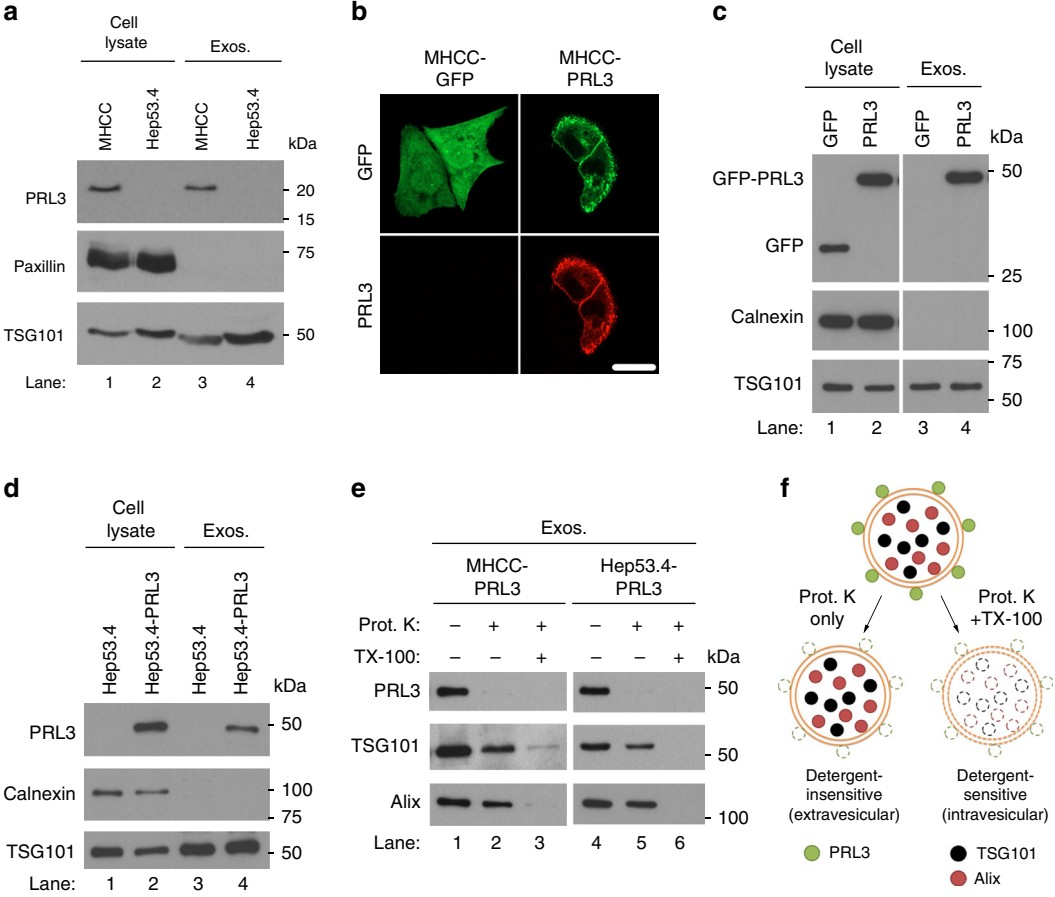

**Fig. 3** PRL3 is expressed on the surface of exosomes from PRL3+ cancer cells. **a** Representative PRL3 western blot (WB) in purified exosomes from MHCC-LM3 and Hep53.4 liver cancer cells cultured for 24 h in serum-free media. Paxillin served as a negative marker for exosomes, whereas TSG101 is a positive exosomal marker. **b** Immunofluorescence analysis of MHCC-LM3 cells transiently transfected with green fluorescence protein (GFP) or GFP-PRL3 to reveal plasma membrane enrichment of PRL3. Scale bar, 20 μm. **c** Representative PRL3 WB in purified exosomes from cells in **b** after culturing for 24 h in serum-free media. Calnexin served as a negative marker for exosomes. **d** Purified exosomes from Hep53.4 and Hep53.4-PRL3 were analyzed as in **c**. **e** Purified exosomes from cells in **c**, **d** were assayed for sensitivity of exosomal PRL3 to proteinase K (Prot. K) digestion in the absence or presence of 1% Triton X-100 detergent. TG101 and Alix are intravesicular exosomal proteins. **f** Proposed model of the localization of PRL3, TSG101, and Alix in exosomes, based on Prot. K susceptibility observed in **e**. Source data are provided as a Source Data file

MHCC-PRL3 cells, the latter clearly demonstrating plasma membrane localization of PRL3 (Fig. 3b). We found that unlike free GFP, which could not be detected in MHCC-GFP-derived exosomes (Fig. 3c, lane 3), GFP-tagged PRL3 robustly accumulated in exosomes secreted by MHCC-PRL3 cells (Fig. 3c, lane 4). Exosome enrichment was validated here using the endoplasmic reticulum-anchored protein calnexin (exosome-negative) and TSG101 (Fig. 3c). We repeated these studies in Hep53.4 and Hep53.4-PRL3 cells and reproducibly demonstrated that PRL3 was not detected in exosomes derived from Hep53.4 parental cells (Fig. 3d, lane 3), but present in exosomes derived from Hep53.4-PRL3 cells (Fig. 3d, lane 4). To test the specificity of PRL3 localization to exosomes, we compared several proteins with various known subcellular localizations, including endosomes (TSG101, Alix), ER (calnexin), focal adhesions (paxillin), nucleolus (fibrillarin), nuclear membrane (nucleoporin), and cytosol (actin; Supplementary Fig. 5). Like PRL3, the endosomal proteins tested (TSG101, Alix) robustly accumulated in exosomes (Supplementary Fig. 5). In contrast, except for actin (Supplementary Fig. 5), none of the other proteins tested with varying subcellular localizations accumulated in exosomal fractions. It should be noted that the presence of actin in exosomes was not

due to its cytosolic localization, as we observed that cytosolic GFP (Fig. 3b) did not localize to exosomes at all (Fig. 3c, lane 3). Thus, PRL3 possesses specific localization signal(s) driving its exosomal localization.

Recently, a proteomic study found that ~33% of 410 exosomal membrane-bound proteins assayed possessed an unconventional flipped "inside-out" protein topology, with typically intracytoplasmic-facing regions of these membrane-bound cellular proteins somehow exposed on the outside of secreted exosomes[28]. To study if membrane-anchored PRL3 might also share such an unconventional inside-out topology to present PRL3 on the outer exosome surface, we employed a protease-protection assay using proteinase K, a broad-acting protease impermeable to lipid bilayers[29,30]. We found that GFP-PRL3 was readily digested by proteinase K (Fig. 3e, lanes 2 and 5), implicating a predominantly extracellular, membrane-associated PRL3 localization on exosomes. In contrast, the classical exosome marker proteins TSG101 and Alix were relatively resistant to proteinase K (Fig. 3e, lanes 2 and 5) and required disruption of the exosomal double membrane with 1% Triton X-100 detergent for comparable sensitivity to proteolytic digestion (Fig. 3e, lanes 3 and 6). These results suggest an intravesicular localization for

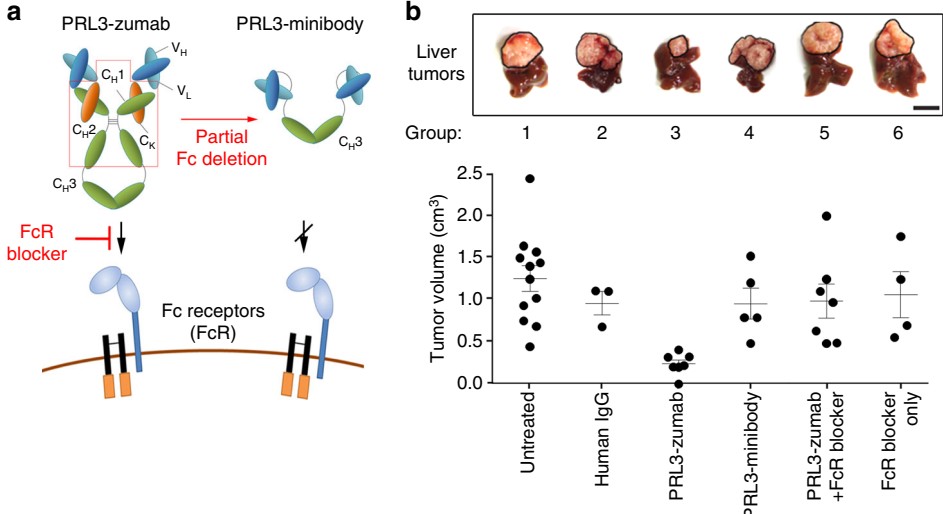

**Fig. 4** PRL3-zumab eliminates tumors in an Fc- and FcR-dependent manner. **a** Cartoon depicting domain architecture of PRL3-zumab (intact Fc) vs. PRL3-minibody (truncated Fc lacking CH1 and CH2 domains) and their ability to engage Fc receptors (FcR) on host immune cells. The anti-CD16/32 FcR blocker antibody prevents IgG from binding murine FcγII/III receptors. **b** Both Fc region of PRL3-zumab and FcR binding are required for anti-tumor effects of PRL3-zumab. Upper panel, representative images of livers from each treatment group at day 35 (5-week endpoint). Tumor areas are framed with black lines. Scale bar, 10 mm. Lower panel, tumor volumes in each group. The mean tumor volumes were calculated using one-way analysis of variance (ANOVA) (mean ± s.e.m.) for independent groups of untreated ($n = 12$), human IgG-treated ($n = 3$), PRL3-zumab-treated ($n = 6$), PRL3-minibody-treated ($n = 5$), PRL3-zumab + FcR blocked-treated ($n = 7$), and FcR blocker-treated ($n = 4$) mice. Source data are provided as a Source Data file

these proteins, consistent with previous reports[28]. Based on these findings, we propose a model for the localization of PRL3 to the surface of exosomes vs. TSG101 and Alix, which are mainly localized within the exosome lumen (Fig. 3f). Collectively, these observations demonstrate that (1) PRL3 possesses specific localization signal(s) for unconventional externalization via exosomes, and that (2) exosomal PRL3 could be a key transport route for "inside-out" PRL3 to the cell surface via membrane fusion.

**PRL3-zumab therapeutic efficacy requires its Fc domain.** Since intracellular PRL3 could be externalized via several unconventional protein-releasing pathways (including exosome secretion), we reasoned that PRL3-zumab might mechanistically exploit tumor clearance pathways similar to conventional antibody therapy targeting other extracellular or surface oncoproteins. To evaluate this, we first investigated the requirement of the Fc region of PRL3-zumab for therapeutic efficacy. Fc receptors (FcRs) on immune cells bind to the constant region (Fc) of IgGs in antigen–antibody complexes and bridge the antibody–antigen complex to host effector cells, such as B lymphocytes, natural killer cells, and macrophages. This results in their recruitment and activation of effector pathways for target antigen/cell clearance via antibody-dependent cell-mediated cytotoxicity (ADCC) or phagocytosis (ADCP)[31]. To investigate the involvement of host FcRs in PRL3-zumab's mechanism of action (MOA), two complementary experiments were designed. First, CH1 and CH2 domains were deleted from the original PRL3 antibody to create a (scFv-CH3)₂ PRL3-minibody with a truncated Fc portion (Fig. 4a, upper cartoon), which still retained PRL3-zumab's variable regions for specific binding to PRL3, but not to its two homologs, PRL1 and PRL2 (Supplementary Fig. 6a, 6b). Second, we co-treated mice with anti-CD16/32 antibodies (Fig. 4a, lower cartoon), which bind to FcγII and FcγIII receptors and inhibit FcR-mediated immune clearance ("FcR blocker")[32].

As the Fc portion of IgG is essential for binding to FcRs[33,34], we evaluated the therapeutic efficacies of full-length PRL3-zumab vs. partial Fc-deleted PRL3-minibody. We generated mice bearing PRL3+ MHCC-LM3 orthotopic liver tumors and divided them into untreated (group I), nonspecific human IgG treatment (group II), PRL3-zumab treatment (group III), PRL3-minibody treatment (group IV), PRL3-zumab co-treatment with FcR blocker (group V), and FcR blocker alone (group VI). Our results showed that compared to untreated tumors (group I), only PRL3-zumab treatment (group III) resulted in significant tumor suppression, whereas the other treatments (groups II, IV, V, and VI) lacked anti-tumor efficacy (Fig. 4b). We further reproduced the lack of PRL3-minibody therapeutic efficacy in a different orthotopic gastric tumor model using the PRL3+ SNU-484 human gastric cell line (Supplementary Fig. 6c), illustrating that this lack of therapeutic efficacy was not an organ-specific defect. Since deletion of PRL3-zumab's CH1 and CH2 domains did not affect the resulting PRL3-minibody's binding to PRL3 (Supplementary Fig. 6a and b), we reasoned that the loss of therapeutic effect was not due to potential antigen-binding defects, but rather due to the lack of FcR binding ability. Taken together with the abolishment of PRL3-zumab therapeutic efficacy upon blockage of host FcγII/III receptors, our results establish that the interaction between the Fc domain of PRL3-zumab and host FcγII/III receptors are essential for anti-tumor therapeutic effects of PRL3-zumab.

**PRL3-zumab recruits immune cells to PRL3+ tumor niches.** The finding that PRL3-zumab required Fc-FcR interaction for in vivo anti-tumor activity indicated the involvement of the classical ADCC pathway of immune-mediated tumor clearance. We thus performed in situ immunofluorescence analysis with antibodies specific to B cells (B220/CD45R) and NK cells (CD335) on PRL3+ MHCC-LM3 orthotopic liver tumor sections derived from untreated mice (group I), PRL3-zumab monotherapy (group II), FcR blocker monotherapy (group III), or

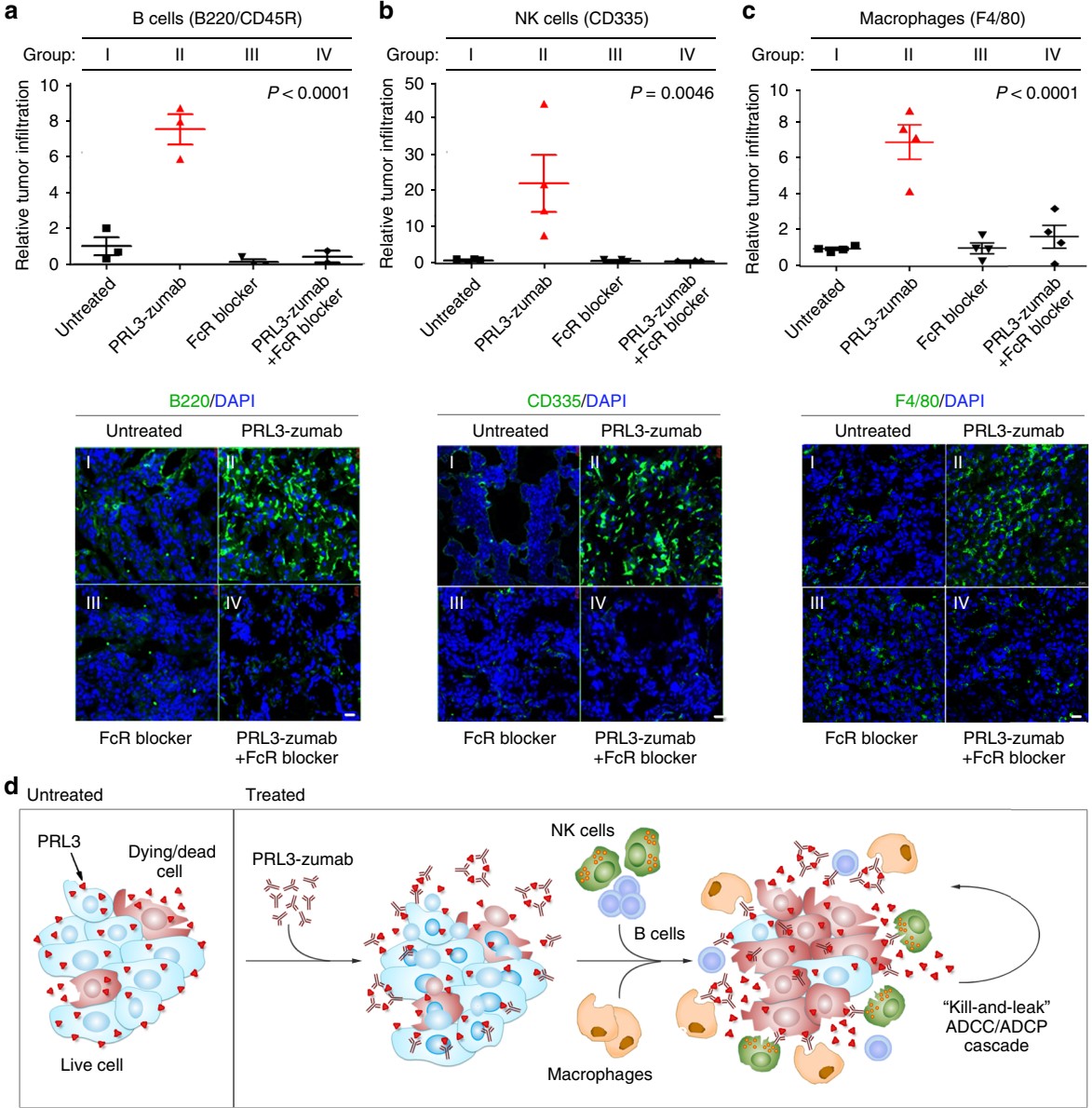

**Fig. 5** PRL3-zumab recruits immune cells to tumor sites in an FcR-dependent manner. **a–c** Orthotopic MHCC-LM3 liver tumor tissue cryo-sections from various groups of mice: I) untreated (filled black squares), II) PRL3-zumab treatment (filled red triangles), III) FcR blocker treatment (filled black triangles), and IV) PRL3-zumab plus Fc blocker combination treatment (filled black diamonds) were analyzed by immunofluorescence with antibodies against B220/CD45R (B cells; **a**), CD335 (NK cells; **b**), or F4/80 (macrophages; **c**) and scored for relative tumor infiltration. The mean relative infiltration was calculated using one-way analysis of variance (ANOVA) (mean ± s.e.m., $n = 4$ independent samples). Lower panels, representative immunofluorescence results for each antibody set. Scale bar, 200 μm. **d** Proposed mechanism of action of PRL3-zumab. Externalized PRL3 antigens are recognized by PRL3-zumab, which then recruits NK cells, B cells, and Ly-6C$^+$F4/80$^+$ macrophages for tumor killing and leakage of additional PRL3 antigens ("kill-and-leak" cycle) triggering an anti-tumor ADCC/ADCP cascade. Source data are provided as a Source Data file

PRL3-zumab+ FcR blocker combination therapy (group IV). In PRL3-zumab-treated tumors (group II), we found a significant enrichment of both B cells (Fig. 5a) and NK cells (Fig. 5b), a phenomenon not observed in untreated and FcR blocker mono-therapy groups (Fig. 5a, b, groups I and III, respectively). Importantly, FcR blocker co-treatment abolished PRL3-zumab-induced accumulation of both B cells and NK cells (Fig. 5a, b, group IV). Whereas NK cells are the major ADCC effectors, macrophage-dependent ADCP is increasingly recognized as the MOA behind many antibodies previously approved to treat cancer[35]. Previous studies have demonstrated that CD45$^+$Lin$^-$CD11b$^+$Ly-6C$^+$ tumor-associated myeloid cells, which include

macrophages, can express F4/80 antigen[36]. Interestingly, significantly higher F4/80$^+$ cell recruitment was evident specifically in PRL3-zumab-treated tumors (Fig. 5c). Using a myeloid immunoprofiling panel with a sequential gating strategy (Supplementary Fig. 7a), we validated the accumulation of tumor-infiltrating CD45$^+$Lin$^-$CD11b$^+$ Ly-6G$^-$Ly-6C$^+$F4/80$^+$ macrophages in PRL3-zumab-treated tumors (Supplementary Fig. 7b, c). In these tumors, we also noted an enrichment of Ly-6C$^+$F4/80$^{low}$ myeloid cells (Supplementary Fig. 7d), a population that typically comprises of phenotypically similar, but functionally distinct, monocytes and/or monocytic myeloid derived suppressor cells (M-MDSCs). Unlike monocytes, which can function

as immune effectors of anti-tumor antibodies via ADCP, M-MDSCs are primarily a pro-tumorigenic myeloid cell population implicated in promoting tumor growth, angiogenesis, and metastasis[37]. As PRL3-zumab therapy is Fc dependent (Fig. 4b), we logically reason that PRL3-zumab more likely recruits tumoricidal monocytes (rather than M-MDSCs) into tumor areas to inhibit (rather than promote) tumor growth. In addition, CD86, an activation receptor found on several immune cell types, including B cells, NK cells, and monocytes/macrophages[38,39], was also more highly expressed by tumor-infiltrating cells in PRL3-zumab-treated tumor sections (Supplementary Fig. 7e, f). Collectively, our results suggest that PRL3-zumab promotes the accumulation and activation of B cells, NK cells, Ly-6C$^+$F4/80$^+$ macrophages, and possibly monocytes, within the tumor microenvironment in an FcR-dependent manner.

Herein, we propose an MOA for PRL3-zumab in Fig. 5d, where the interaction between the Fc domain of PRL3-zumab and FcγII/III receptors on host immune cells results in their recruitment to tumors expressing externalized PRL3 antigens to activate classical antibody-mediated tumor clearance pathways in vivo, triggering an ADCC/ADCP feedforward cascade encompassing cyclical cell killing and PRL3 leakage ("kill-and-leak" ADCC/ADCP cascade).

**PRL3 is frequently overexpressed in multiple human cancers.** We previously demonstrated the value of PRL3 as a gastric cancer oncotarget, where PRL3 expression was detected in approximately 85% of fresh-frozen gastric tumor tissues, but not in patient-matched normal gastric tissues[11]. Since elevated *PRL3* transcript expression has been described in many other tumor types[40], we sought to broadly characterize PRL3 protein expression in 151 hard-to-obtain, fresh-frozen patient tumor samples from 11 different cancer types, including several aggressive malignancies with unmet medical needs. In these randomly allocated fresh-frozen samples sourced mainly from the Tissue Repository of the National University Hospital Singapore and other clinical collaborators, we detected high PRL3 expression in 16/20 liver tumors (80%; Fig. 6a), 9/10 lung tumors (90%; Fig. 6b), 7/10 colon tumors (70%; Fig. 6c), 9/10 breast tumors (90%; Fig. 6d), 12/14 stomach tumors (86%; Fig. 6e), 11/12 thyroid tumors (92%; Fig. 6f), 6/7 pancreatic tumors (86%; Fig. 6g), 13/18 kidney tumors (72%; Fig. 6h), 6/12 acute myeloid leukemia (AML) bone marrow samples (50%; Fig. 6i), 24/34 bladder tumors (71%; Fig. 6j), and 4/4 prostate tumors (100%; Fig. 6k). For most of the liver, lung, colon, breast, pancreatic, thyroid, and kidney tumors, as well as one bladder tumor sample, we further managed to obtain fresh-frozen, patient-matched non-cancerous adjacent tissue pairs from the same organs, which allowed precious insight into tumor-specific PRL3 expression. Herein, PRL3 was not detected in any of the matched normal tissue types examined, despite high expression in corresponding matched tumors (Fig. 6a–h, j).

Our previous reports demonstrate that PRL3 antibody therapy can inhibit PRL3$^+$ tumors in multiple animal models using cancer cell lines derived from different origins, including ovarian, colorectal, melanoma, lung, AML, and gastric tissues (Supplementary Table 1). As our results here demonstrate that PRL3 is a broad, tumor-associated target expressed in an average of 80.6% of human tumors across 11 varied tumor types (Table 1), a question that remained was whether PRL3$^+$ human tumors, like mouse tumors (Fig. 2f), also express surface PRL3. To answer this, we studied the relationship between total cellular PRL3 expression and surface PRL3 expression using a freshly resected matched pair of normal and tumor kidney tissues from a renal cancer patient. Pathological evaluation using hematoxylin and eosin staining confirmed the absence of tumor cells in the normal sample, while >90% of the tumor sample demonstrated salient features of clear cell renal carcinoma, without any normal kidney parenchyma notable (Supplementary Fig. 8a). Western blotting of tissues lysates showed that PRL3 was not detected in the normal kidney sample, but strongly expressed in the kidney tumor sample (Supplementary Fig. 8b), in line with PRL3 protein as a tumor-specific target. Consistently, PRL3 surface+ cell populations could be detected only in dissociated PRL3$^+$ tumor cells, but not in PRL3$^-$ normal cells (Supplementary Fig. 8c and 8d). We also detected PRL-3 surface+ tumor cell populations in a freshly resected nasal tumor sample (Supplementary Fig. 8e and 8f).

**Discussion**

In this study, we present five key findings: (1) generation of an orthotopic "seed and soil" liver tumor model using the human PRL3$^+$ HCC cell line, MHCC-LM3, as a clinically relevant liver tumor model for PRL3-zumab therapeutic evaluation, (2) externalization of PRL3 to the cell surface of tumor isolates and serum-starved cultured cells as a direct surface target for PRL3-zumab, (3) exosomal secretion of PRL3 as a possible pathway for intracellular PRL3 to become externalized as surface PRL3, (4) identification of the essential role for the Fc region of PRL3-zumab to bind FcRs and recruit host immune cells for anti-tumor efficacy, akin to classical mechanisms for antibody therapy, and (5) the clinical value of PRL3 as a frequently expressed oncotarget in 11 common cancer types.

Our finding that PRL3-zumab effectively suppressed PRL3$^+$ orthotopic liver tumors presents a solution to a long-standing challenge in medical treatment of HCC, whose pathophysiologic complexity is often exacerbated by underlying functional liver insufficiency. For instance, sorafenib, the first targeted therapy for advanced HCC that demonstrated a slight improvement in mortality[41], actually resulted in worse survival outcomes in patients with both advanced HCC and liver dysfunction (Child-Pugh B patients) due to toxicity[42]. Given that at least five major phase 3 trials of molecular-targeted agents against advanced liver cancer have failed in the past decade[43], the frequent overexpression of PRL3 in liver cancer patients (80%; Fig. 6a) highlights PRL3-zumab as a potentially valuable therapeutic option for HCC by fulfilling the unmet need for efficacious and well-tolerated targeted drugs to treat this morbid disease.

Using the MHCC-LM3 orthotopic liver tumor model, we consolidated the MOA for PRL3-zumab by providing key evidence demonstrating how intracellular PRL3 oncoprotein can be externalized as surface PRL3 for PRL3-zumab to trigger the host immune system's canonical pathways of antibody-mediated tumor clearance within tumor microenvironment. It should be highlighted that PRL3 is not the only intracellular protein to be externalized by malignant cells. HSP70, HSP90, GRP78, actin, cytokeratins, vimentin, and feto-acinar pancreatic protein are all related examples of intracellular proteins in normal cells, which are externalized by cancerous cells[24]. Early studies have also documented the generally higher permeability of tumor cell membranes as compared to normal cells[44]. Although surface PRL3 could arise by several possible pathways, we found that intracellular, lipid-anchored PRL3 could be externalized on the surface of exosomes with an "inside-out" topology. Notably, most lipid-anchored Rab family proteins have also been predicated as having such a topology on exosomes[28], an observation experimentally validated for Rab5[45]. Regardless of the route employed, we propose that the externalization of PRL3 in tumors makes it possible to selectively target this oncoprotein with antibody therapy in the same manner as conventional targeting of classical cell surface or secreted targets, ultimately resulting in feedforward "kill-and-leak" cascades facilitating tumor elimination (Fig. 5d).

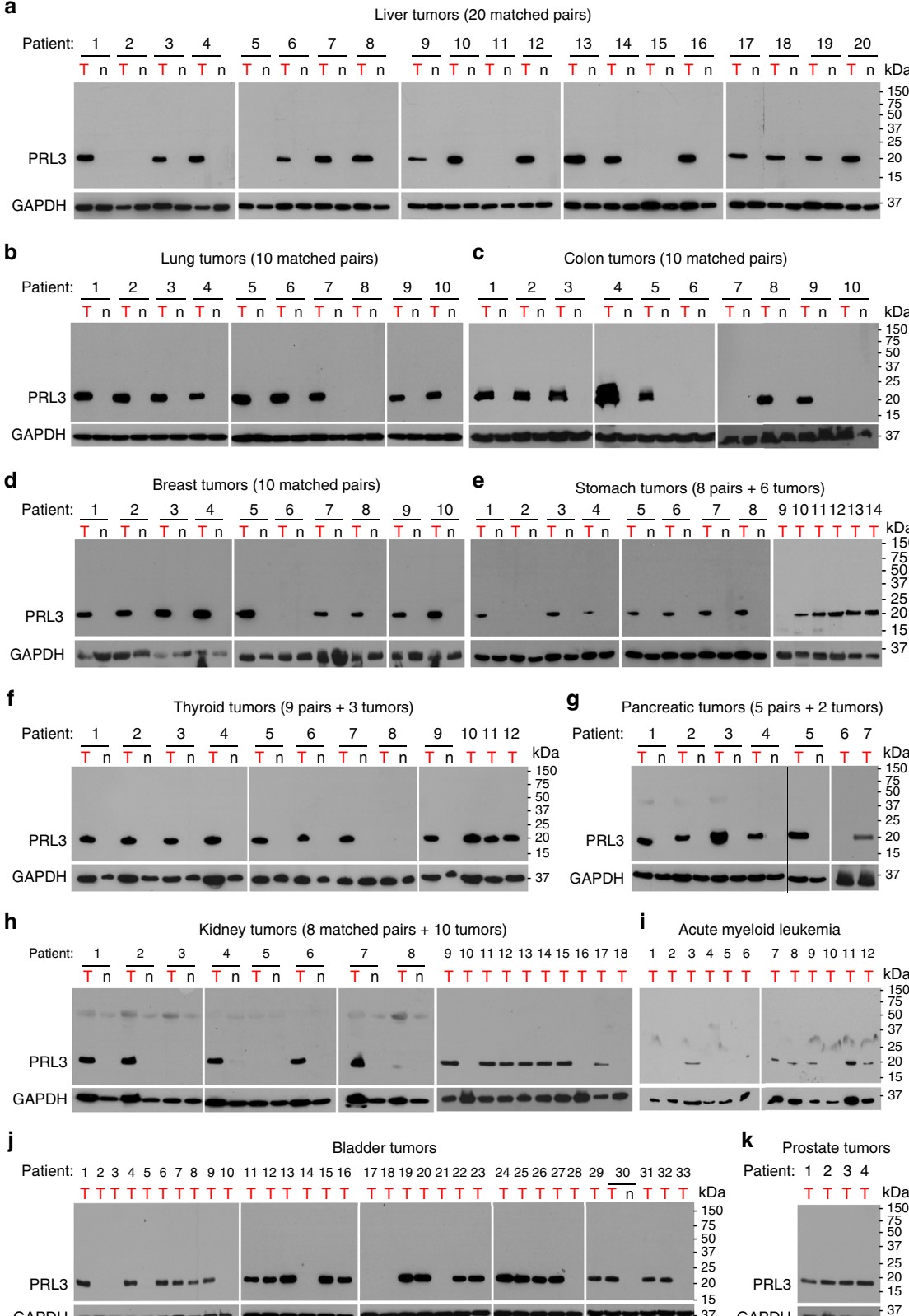

**Fig. 6** PRL3 is frequently overexpressed in multiple human cancers. **a–k** Representative full western blot (WB) of PRL3 protein levels in tumor ("T") tissues and, where available, patient-matched normal ("n") tissues from **a** liver, **b** lung, **c** colon, **d** breast, **e** stomach, **f** thyroid, **g** pancreas, **h** kidney, **i** acute myeloid leukemia (bone marrow aspirates), **j** bladder, and **k** prostate tissues. Relative molecular masses (in kDa) are indicated on the right of each immunoblot. Glyceraldehyde 3-phosphate dehydrogenase (GAPDH) served as a loading control. Source data are provided as a Source Data file

**Table 1 Summary of PRL3 expression across different tumor types**

| Tumor type | PRL3+ | PRL3− | Total | % PRL3+ |
|---|---|---|---|---|
| Liver | 16 | 4 | 20 | 80 |
| Lung | 9 | 1 | 10 | 90 |
| Colon | 7 | 3 | 10 | 70 |
| Breast | 9 | 1 | 10 | 90 |
| Stomach | 12 | 2 | 14 | 86 |
| Thyroid | 11 | 1 | 12 | 92 |
| Pancreas | 6 | 1 | 7 | 86 |
| Kidney | 13 | 5 | 18 | 72 |
| AML | 6 | 6 | 12 | 50 |
| Bladder | 24 | 10 | 34 | 71 |
| Prostate | 4 | 0 | 4 | 100 |
| Total | 117 | 34 | 151 | |
| Average | | | | 80.6 |

*PRL3* phosphatase of regenerating liver 3

Despite potent anti-tumor activity in vivo, PRL3-zumab did not result in any killing or suppression of PRL3+ cancer cell growth in vitro. Our findings herein support three possible explanations for this discrepancy. First, the proportion of PRL3 surface+ cells is much lower in cultured cells compared to tumor cells, likely failing to reach sufficient threshold levels to trigger PRL3-zumab-mediated inhibitory effects in vitro. Second, in vitro artificial culture conditions fail to recapitulate complex host factors within the tumor microenvironment. True to this, we found that serum starvation, which mimics the deprivation of growth-promoting factors common in poorly vascularized tumors, enhanced the presence of surface PRL3 surface on cancer cells. Third, complex tumor–host immunity interactions are unable to be faithfully recaptured using in vitro systems. This is a major limitation, as we found that in vivo Fc-FcγR binding is essential for anti-tumor effects of PRL3-zumab by recruiting immune cells into PRL3+ tumor microenvironments. In tumor sections, PRL3-zumab promoted tumoral infiltration of B cells, NK cells, and macrophages, which are important mediators of ADCC and ADCP. Intriguingly, this sets PRL3-zumab apart from other intracellular-targeting antibodies, which typically rely on cell penetration and target inhibition[46–48], and instead places its MOA together with well-established cell surface-targeting drugs such as trastuzumab and rituximab. Our findings herein demonstrate that the in vivo environment plays a crucial role in influencing the druggability of target proteins and their therapeutic responses, and that in vivo drug screening should be employed for drug development studies to circumvent limitations of assays based on simplified laboratory assays.

PRL3 is specifically overexpressed in 80.6% of randomly analyzed human tumors (across 11 cancer types), but not in any matched normal tissues examined, making it an attractive general tumor antigen to treat many human cancers. Moreover, PRL3-zumab appears well tolerated by cancer patients in an ongoing phase 1 trial in Singapore (https://clinicaltrials.gov/ct2/show/NCT03191682?term=PRL-3&rank=1). Such a safety profile, coupled with the preclinical efficacy of PRL3-zumab in orthotopic tumor models, lends support for further studies of PRL3-zumab in PRL3+ cancer patients as a safe, effective targeted therapy.

## Methods

**Cell culture.** MHCC-LM3 human liver cells were provided by Prof. Kam Man-Hui from the National Cancer Center (Singapore). Hep3B2.1 (HB-8064), HepG2 (HB-8065), and PLC (CRL-8024) liver cancer cell lines were sourced from the American Type Tissue Collection (ATCC). Huh-7 human liver cancer cells (JCRB0403) were sourced from the Japanese Collection of Research Bioresources (JCRB) Cell Bank. SNU449 human liver cancer cells (#00449) were sourced from the Korean Cell Line Bank (KCLB). Hep53.4 murine liver cancer cells (CLS 400200) were sourced from Cell Line Service GmbH. All liver cancer cells were maintained in Dulbecco's modified Eagle's medium (Gibco) supplemented with 10% (v/v) fetal bovine serum (FBS; Hyclone) and 2 mM L-glutamine. SNU-484 gastric cancer cells (#00484; KCLB), B16F0 mouse melanoma cells (CRL6322; ATCC), and CHO-K1 ovarian cells (CCL61; ATCC) were grown in RPMI-1640 medium (Gibco) with the same supplements as above. All cells were grown at 37 °C in a humidified incubator under 5% CO₂ atmosphere and tested for mycoplasma contamination using the EZ-PCR mycoplasma test kit (Biological Industries). The Hep53.4-PRL3 cell line, which stably expresses EGFP-tagged PRL3 fusion protein, was generated by using jetPRIME reagent (Polyplus-transfection) to transfect Hep53.4 cells with the pEGFP-C1-PRL-3 plasmid[49]. Stable cell clones after selection for 2 weeks with 1 mg mL−1 G418 (Clontech) were pooled and used for subsequent studies.

**Antibodies.** Murine anti-PRL3 monoclonal antibody (mAb) (clone 318) was generated in-house[49]. PRL3-zumab was engineered based on the original framework of murine anti-PRL3 mAb (clone 318) and extensively validated for specific PRL3 binding[11]. GAPDH (clone MAB374) antibody was purchased from Millipore. GFP (clone B-2) and actin (clone H-196) antibodies were purchased from Santa Cruz Biotechnology. TSG101 (14497-1-AP) antibody was purchased from Proteintech. Alix (clone 3A9) and fibrillarin (clone C13C3) antibodies were purchased from Cell Signaling Technologies. Calnexin (cat# 610524), nucleoporin p62 (cat# 610497), and paxillin (cat# 610051) antibodies were purchased from BD Biosciences. Horse radish peroxidase (HRP)-conjugated goat anti-mouse, anti-rabbit, and anti-human (H + L) secondary antibodies and fluorescein isothiocyanate (FITC)-conjugated goat anti-mouse antibody were purchased from Jackson ImmunoResearch. FITC-conjugated goat anti-human antibody was purchased from Life Technologies. CD335/Nkp46 (clone 29A1.4), B220/CD45R (clone RA3-6B2), and CD86 (clone GL1) were purchased from BD Pharmingen. F4/80 antibody (clone REA126) was purchased from Miltenyi Biotec. Cetuximab was purchased from Zuellig Pharma Asia Pacific. Anti-CD16/CD32 antibody (clone 2.4G2), polyclonal human IgG, and polyclonal mouse IgG were purchased from Bio X Cell. Additional antibody details and working dilutions used for various applications are provided in Supplementary Table 2.

**Preparation of tissue and cell lysates.** Excised tissue samples (5 mm³) were suspended in RIPA lysis buffer (150 mM NaCl, 1.0% IGEPAL CA-630, 0.5% sodium deoxycholate, 0.1% sodium dodecyl sulfate (SDS), 50 mM Tris, pH 8.0) supplemented with a protease and phosphatase inhibitor cocktail (Roche) for 15 min at 4 °C and disrupted completely with a tissue homogenizer (Polytron). Lysates were clarified by centrifugation at 13,000 × g for 40 min at 4 °C. For cultured cells, 5 × 10⁶ cells were lysed in lysis buffer and clarified as described above. Protein concentrations of both tissue and cell lysates were estimated using a bicinchoninic assay kit (Pierce). After the addition of 2× Laemmli buffer containing dithiothreitol (50 mM final concentration), samples were boiled and used immediately for Western blotting or stored at −80 °C until use.

**Isolation of exosomes.** Exponentially growing cells in T-75 flasks were cultured in complete media till 70–80% confluence and washed twice with phosphate buffered saline (PBS) before incubation in serum-free media (10 mL per flask) for 24 h. For transient transfection studies, 10 μg of pEGFP-C1 or pEGFP-C1-PRL-3 plasmids were pre-mixed with 25 μL of jetPRIME reagent and added to exponentially growing MHCC cells (3 × 10⁶) pre-seeded in T-75 flasks for 24 h, before washing with PBS and incubation in serum-free media for another 24 h. We used serum-free media in these cellular studies to avoid the contamination of bovine exosomes abundantly present in FBS. The harvested culture medium was first subjected to serial centrifugation at 300 × g for 5 min to remove floating cells and at 2000 × g for 20 min to remove cell debris. The clarified culture medium (~50–100 mL from 5 to 10 flasks) was then concentrated to ~250 μL using Amicon Ultra-15 centrifugal filters (100 K MWCO; EMD Millipore), mixed thoroughly with 0.5 volumes of Total Exosome Isolation Reagent (Thermo Fisher Scientific), and incubated for 16 h at 4 °C. Exosomes were finally pelleted by centrifugation at 10,000 × g for 60 min at 4 °C, resuspended in PBS, and analyzed immediately or stored at −20 °C till use.

**Proteinase K treatment.** Exosomes (20 μg) resuspended in PBS were incubated with 2 μg ml−1 Proteinase K (Invitrogen) in the presence of 5 mM CaCl₂ for 20 min at 37 °C. The proteinase activity was then inhibited by adding 5 mM phenylmethylsulfonyl fluoride for 5 min on ice and immediately analyzed.

**Western blotting.** Tissue, cellular, and exosomal lysates were resolved on SDS-polyacrylamide gels and transferred to nitrocellulose membranes before incubation with blocking buffer (5% skim milk in 20 mM Tris, pH 7.6, 140 mM NaCl, 0.2% Tween-20) for 1 h, followed by overnight incubation at 4 °C with primary antibodies diluted in blocking buffer (working dilutions given in Supplementary Table 2). After thorough washing with TBS-T buffer (20 mM Tris, pH 7.6, 140 mM NaCl, 0.2% Tween-20), membranes were incubated with HRP-conjugated secondary antibodies for 1 h, thoroughly washed with TBS-T, and visualized using a

chemiluminescent substrate (Millipore). Uncropped and unprocessed scans of all blots are provided in the Source Data file.

**Mouse tumor models and treatments**. Eight-week-old male NCr nude mice (InVivos Pte Ltd, Singapore) were used for all animal models in this study. Mice were anesthetized intraperitoneally with a cocktail comprising ketamine (150 mg kg$^{-1}$) and xylazine (10 mg kg$^{-1}$). Abdomens of anesthetized mice were opened in layers by a 1-cm midline incision starting just below the xiphoid sternum. For orthotopic liver tumor models, the liver was exposed and HCC cells ($3 \times 10^6$ for MHCC-LM3, $5 \times 10^5$ for Hep53.4 and Hep53.4-PRL3) were inoculated in a total volume of 50 µL into the subcapsular layer of the left lobe of livers. For orthotopic gastric tumor models, $3 \times 10^6$ SNU-484 cells were inoculated in a total volume of 50 µL into the subserosa layer of the exposed stomach. The abdominal wall was then sutured back in layers. Because of different growth rate of individual tumors, the duration of experiments were 5 weeks (35 days) for MHCC-LM3 tumors, 4 weeks (28 days) for SNU-484 tumors, and 3 weeks (21 days) for Hep53.4 and Hep53.4-PRL3 tumors. The treatment regime commenced on day 5 post inoculation of MHCC-LM3 cells, or day 2 post inoculation for SNU-484, Hep53.4, and Hep53.4-PRL3 cells. For tumor growth/volume experiments, treated mice were administered intravenously (i.v.) twice a week with 100 µg each of either PRL3-zumab, polyclonal human IgG (Bio X Cell), PRL3-minibody, or anti-CD16/32 antibody (clone 2.4G2; Bio X Cell). Co-treatment was performed by co-administration of 100 µg each of PRL3-zumab and anti-CD16/32 antibody. All antibodies were diluted into 100 µL (final) of PBS for injection. Untreated mice were administered i.v. with an equivalent volume of placebo (buffer alone) as a control. Final tumor volumes were calculated using the formula: volume = $0.4 \times$ tumor length × (tumor width × tumor width). For survival studies, treated mice were administered i.v. with 100 µg of PRL3-zumab twice a week for a total of 10 times. All animals were closely monitored 3 times per week throughout the duration of experiment. For survival studies, the endpoint chosen adhered to guidelines stipulated by the A*STAR Institutional Animal Care and Use Committee (IACUC), which recommends that animals be euthanized (and recorded as a "death" event in our survival analysis) upon tumor-induced abnormal behavior, emaciated appearance, or moribund condition. For tail vein metastasis models, 8-week-old wild-type C57BL/6 mice (InVivos Pte Ltd., Singapore) were injected with $1 \times 10^6$ B16F0 mouse melanoma cells intravenously via the lateral tail vein. Body weight and general activity of mice were monitored. After 3 weeks, mice were euthanized and metastatic tumors identified as black-colored nodules in the lung. Animal studies were approved by the A*STAR Institutional Animal Care and Use Committee (IACUC) and performed in accordance with approved guidelines and regulations.

**Cell viability assay**. Cell viabilities were assessed using the MTS (3-(4,5-dimethylthiazol-2-yl)−5-(3-carboxymethoxyphenyl)−2-(4-sulfophenyl)-2H-tetrazolium)-based CellTiter 96 AQueous One Solution Cell Proliferation Assay (Promega) according to the manufacturer's instructions. Briefly, $2 \times 10^3$ cells were seeded in complete media into triplicate wells of a 96-well plate and allowed to attach overnight. The medium was then replaced with complete media containing PBS (0.1%; control), PRL3-zumab (5 or 50 µg mL$^{-1}$), or cisplatin (Hospira UK; 2 or 10 µg mL$^{-1}$) and left to incubate for another 48 h at 37 °C in 5% CO$_2$ atmosphere. The media were subsequently aspirated and replaced with 150 µL fresh media containing MTS (Promega) and formazan development was done for 2 h at 37 °C and 5% CO$_2$ before measuring absorbance at 490 nm in a spectrophotometer (Tecan).

**Cell surface analysis**. Freshly harvested orthotopic MHCC-LM3 liver tumors and human kidney normal/tumor tissue samples were removed of fat and fibrous areas before being cut into 2–4 mm pieces for dissociation using a human Tumor Dissociation Kit and gentle MACS Octo Dissociator (Miltenyi Biotec) following the manufacturer's recommended protocol for dissociation for "tough" tissues such as liver and kidney. This kit is optimized for high cell yield of tumor cells, while preserving important cell surface epitopes. Dissociated cells were subsequently counted, resuspended in RPMI, and kept on ice till analysis. For analysis of cultured cells in vitro, exponentially growing cells at 80% confluence in T-75 flasks were washed once with PBS and incubated with 2 mL non-enzymatic cell dissociation buffer (Sigma-Aldrich) for 5 min to dislodge the adherent cells into suspension. In serum starvation experiments, cells were washed twice with PBS and incubated with serum-free RPMI at 37 °C and 5% CO$_2$ for the indicated durations prior to harvest. Harvested cells were washed once with PBS, counted, resuspended in full RPMI media, and kept on ice till analysis. For cell surface analysis by flow cytometry, $4 \times 10^5$ dissociated cells were washed with PBS and resuspended in PBS containing 1:1000 dilution of LIVE/DEAD Fixable Near-IR stain (Thermo Fisher Scientific). Cells were then washed and incubated with 2 µg of cetuximab (chimeric anti-EGFR mAb), PRL3-zumab (humanized anti-PRL3 mAb), m318 (murine anti-PRL3 mAb), or polyclonal human or mouse IgG in a total volume of 100 µl binding buffer (PBS supplemented with 2% FBS) for 30 min at 4 °C. After incubation, 1 mL binding buffer was added to each sample, centrifuged, and the cell pellet resuspended in 100 µL binding buffer containing 1.5 µL of FITC-conjugated secondary antibody. After a 15 min incubation at

4 °C, the cells were washed twice with 1 mL Annexin Binding Buffer (ABB; 10 mm HEPES, pH 7.4, 150 mm NaCl, 2.5 mm CaCl$_2$) and resuspended in 100 µL ABB containing 2.5 µL Annexin-V PE (BD Biosciences) or Annexin-V Alexa Fluor 350 (Thermo Fisher Scientific). After a 15 min incubation at 4 °C, cells were washed once with ABB and finally resuspended in 150 µL ABB prior to analysis. All samples were run on a BD LSR II flow cytometer and analyzed using the Flowing Software 2 (Turku Center for Biotechnology) or FlowJo software (FlowJo).

**Immunoprofiling of tumor-infiltrating cells**. Single-cell suspensions of tumors isolates were first stained with Zombie UV Fixable Viability dye (BioLegend) for 30 min at 4 °C. Non-specific labeling was blocked with anti-CD16/32 (clone 2.4G2; BD Biosciences) for 30 min at 4 °C before multiplex labeling for 30 min at 4 °C with the following antibodies from BioLegend: AF488 anti-mouse CD45 (clone 30-F11; 1:30 dilution), Brilliant Violet 785 anti-mouse CD3e (clone 145-2C11; 1:30 dilution), Brilliant Violet 785 anti-mouse CD45R/B220 (clone RA3-6B2; 1:60 dilution), Brilliant Violet 421 anti-mouse CD335 (clone 29A1.4; 1:15 dilution), PE/Dazzle 594 anti-mouse CD11b (clone M1/70; 1:60 dilution), APC/Cy7 anti-mouse F4/80 (clone BM8; 1:30 dilution), PE/Cy7 anti-mouse Ly-6G (clone 1A8; 1:60 dilution), and Brilliant Violet 711 anti-mouse Ly-6C (clone HK1.4; 1:60 dilution). Lineage markers (Lin) were defined as CD3, CD335, and B220. All samples were run on a BD LSR II flow cytometer and analyzed using the FlowJo software (FlowJo). The gating strategies used to identify Ly-6C$^+$F4/80$^+$ macrophage and Ly-6C$^+$F4/80$^{low}$ myeloid cell populations are provided in Supplementary Fig. 9.

**Generation of anti-PRL3 (scFv-CH$_3$)$_2$-minibody**. The DNA sequence of PRL3-zumab antibody was sent to ImaginAb (CA) for the synthesis of the anti-PRL3 (scFv-CH$_3$)$_2$-minibody ("PRL3-minibody") using their proprietary in-house methodology. PRL3-minibody is a smaller (~80 kDa) version of PRL3-zumab, devoid of Fc C$_H$1 and C$_H$2 domains but validated to retain specific binding towards the conserved PRL3-zumab epitope (Supplementary Fig. 4a, b).

**Enzyme-linked immunosorbent assay**. Nintery-six-well plates coated overnight with 1 ng of GST-PRL1, GST-PRL2, or GST-PRL3 were blocked with 3% bovine serum albumin (BSA) in PBS containing 0.05% Tween-20 (PBS-T) prior to incubation with 200 ng PRL3-zumab, PRL3-minibody, or human IgG for 2 h at 37 °C. After extensive washing in PBS-T, HRP-conjugated anti-human antibody was added for 1 h at 37 °C. Colorimetric development was performed using a TMB substrate kit (Thermo Fisher Scientific) and stopped by acidification with 2 M H$_2$SO$_4$. Absorbance was measured at 450 nm using a plate reader (Tecan).

**Immunofluorescence analysis**. Fresh-frozen specimens of MHCC-LM3 orthotopic liver tumors were sectioned into 10 µm slices using a cryostat (Leica) at 16 °C and transferred onto poly-L-lysine-coated slides (VWR). For cultured cells, cells were seeded onto glass coverslips in complete media 24 h prior to fixing. Slides or coverslips were fixed with 4% paraformaldehyde for 20 min, washed with PBS-0.05% Tween-20, and blocked in PBS-FDB (PBS, pH 7.0, 2% BSA, 5% goat serum, 5% fetal bovine serum) for 1 h at room temperature. These were subsequently incubated with the indicated primary antibodies at a 1:100 dilution 4 °C overnight, washed, and incubated for 2 h with their cognate fluorophore-conjugated secondary antibodies at a 1:200 dilution. After staining with 300 nM 4′,6-diamidino-2-phenylindole (DAPI) (Sigma-Aldrich) in PBS for 3 min, slides or coverslips were mounted in Vectashield anti-fade reagent (Vector Laboratories), and finally sealed using nail polish. Confocal imaging was performed with an LSM800 confocal microscope (Zeiss AG). Three representative images of tumor-infiltrating lymphocytes in the tumor area adjacent to the tumor capsule (junction of normal and tumor tissues) were taken for each tissue section. For each immune marker analyzed, relative tumor infiltration was determined by calculating the average ratio of FITC-positive cells (immune cell marker) to DAPI-stained nuclei in these representative images, and normalizing this value to the average ratio in untreated mice (group I). The results were presented as means ± s.e.m.

**Fresh-frozen multiple cancer patient samples**. One hundred and fifty-one multiple human fresh-frozen tumor samples, 73 of which had corresponding adjacent non-tumor (normal) tissue samples, were obtained from National University of Singapore Tissue Repository (NUHS-TR), National Cancer Center Singapore (NCCS), and Singapore General Hospital (SGH). Fresh, surgically excised tumor samples were obtained from SGH and analyzed within the same day. Written informed consent was obtained from all patients. The collection and use of human tissue samples were approved by the Institutional Review Board of the National University of Singapore and the National Healthcare Group, Singapore. Specimens were collected and stored at liquid nitrogen immediately after surgery. Total proteins were extracted and subjected to Western blot analysis.

**Statistics**. Tumor volumes between untreated and treated mice were analyzed using two-tailed Student's *t* test. Tumor volumes and immune cell infiltrations among multiple treatment groups were separately analyzed using one-way analysis of variance and, where indicated, post hoc Tukey's honest significance test test. The

log-rank test was used to assess significant differences in the Kaplan–Meier analysis of overall survival between untreated and treated mouse groups. GraphPad Prism v4.0 (GraphPad Software) was used for statistical calculations, and *P* values <0.05 were considered statistically significant.

**Reporting summary**. Further information on research design is available in the Nature Research Reporting Summary linked to this article.

## Data availability

All data generated or analyzed during this study are included in this published article (and its supplementary information files). The source data underlying all Main and Supplementary Figures are provided as a Source Data file.

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

## Acknowledgements

This work was supported by research grants from the Agency for Science, Technology and Research (A*STAR), Singapore. We are grateful to Dr. Eng Chon Boon and the National University Hospital Tissue Repository, Singapore, for providing clinical samples, as well as to the Advanced Molecular Pathology Laboratory (IMCB, A*STAR) for pathological analysis and H&E staining of the patient kidney samples. We are appreciative to Prof. John Connolly and Dr. Sriram Narayanan for their support in flow cytometry analysis, and to Dr. Manikandan Lakshmanan and Mr. Anandhkumar Raju

for assistance with clinical samples. We also thank Professor Sir David Lane (Chief Scientist, A*STAR) for assisting with PRL3-minibody generation with ImaginAb, Inc., USA.

## Author contributions

M.T., A.Q.A., and Q.Z. designed the experiments and prepared the manuscript. M.T., A.Q.A., A.G., J.S.P.Y., J.L., J.X.E.S., and N.Y.Z.T. performed the experiments. C.E.C., S.C.L., K.M.H., Y.K.G., W.P.Y., J.S., W.J.C., C.H., J.B.Z., L.Z.W., J.S.P.Y., S.M.Y., H.S.S.H., E.C., S.P.C., J.N., M.C.H.N., C.C., E.S.A.Y., I.B.H.T., J.P.T., and B.C.G. provided materials and analyzed the results. M.T., A.Q.A., J.S.P.Y., and Q.Z. proofread and finalized the manuscript. All authors approved the manuscript.

## Additional information

**Competing interests:** Q.Z. is the founder of Intra-Immu SG Pte Ltd., an Agency of Science, Technology and Research (A*STAR) spin-off company granted licensing rights for the PRL3-zumab IP portfolio. The other authors declare no competing interests.

