## [Peer Review File · Nature Communications]

Reviewers' comments:

Reviewer #1 (Ab cancer therapy)(Remarks to the Author):

The manuscript entitled "PRL3-zumab blocks tumors expressing 'intracellular' PRL-3 as a novel anti-cancer immunotherapy" by Thura et al describes novel antibody that targets Phosphatase of regenerating liver 3 (PRL3). PRL3 provokes a tyrosine phosphoproteome to drive prometastatic signal transduction. PRL3 is suspected to be a causative factor toward cellular metastasis when in excess. PRL3 is associated with primary tumor growth, tumor reoccurrence, and therapy resistance in many human cancers. Consequently it is perceived as an important player. However, PRL3 is localized as an intracellular protein but can be expressed in plasma membrane remodeling.

The authors describe a set of studies in which a first-in-class humanized antibody for cancer therapy, PRL3-zumab, reactive with PRL3 was examined to understand its functional role, why it sees the target and the expression pattern of this protein in a several human cancers. The authors provide very detailed results showing that the tumor microenvironment plays a significant role in the surface expression of the target, in which a mechanism is proposed and that the FC region of the monoclonal regulates the functional properties of the antibody. These are all important observations. The manuscript is well written and the experiments are described in great detail. The results support the conclusions drawn. The statistical treatments of the results are appropriate. The only criticism is that the manuscript is very long. For example the Introduction could be cut back to 3 pages. One of the most important elements of this manuscript is that in vitro assessments are not always "truthful". The relationship between microenvironment stresses leading to surface expression is important in its own right.

Reviewer #2 (Cancer immunity, cancer biology)(Remarks to the Author):

In the present study Min Thura et al. describe their findings regarding the effects of an antibody against PRL-3 on tumor growth in vivo. PRL-3 (a lipid anchored dual-specificity protein phosphatase of regenerating liver-3 gene) is expressed in the cytoplasm of tumor cells and is associated with metastasis in colorectal cancer. The authors found that in tumor cells isolated from tumor bearing mice but not in counterparts of the same tumor cells cultured in vitro, PRL-3 can be detected on the cell surface. They support that the extracellular localization of PRL-3 in the tumor cells is a consequence of exosome formation and can be a therapeutic target for cancer treatment because an antibody against PRL-3 recruited NK cells, B cells and M1 macrophages in the tumor microenvironment by Fc receptor-mediated mechanism and decreased tumor growth. The results are interesting and have therapeutic potential. Several points require attention before conclusions can be made:

Specific points:

1) Figure 2b-d: The right upper quadrant of this figure shows enhanced non-specific binding of control anti-human Ig in PI positive tumor cells which is highly elevated compared to cultured tumor cells. This finding indicates non-specific binding of the antibody in the dying tumor cells.

Subsequently, the investigators assess binding of EGFR antibody and find a similar pattern in the tumor cells. Based on these findings they concluded that the binding is due to non-specific trapping of the antibody by dead cells. For this reason they decided to use PI negative tumor cells (upper left quadrant) for the remaining of the studies. There are several problems with the results shown in this figure:

a) The assessment of PI negative cells in Figure 2c is inaccurate because the numbers and histograms shown in the upper left quadrants of the right and left panels in 2c are incompatible.

b) The tumor cells included in the left upper quadrant of the right hand side panels (labelled 84.4% in

c, and 64.4% in d) is most likely a population of proapoptotic tumor cells on their way to undergo cell death and become PI positive. They authors should assess both annexin V and PI and exclude apoptotic and dead cells.

Rapidly proliferating tumor cells undergo cell death and this is likely the reason for which become positive for "surface" expression of PRL-3. The authors should assess proapoptotic, apoptotic and dead cells in tumors isolated from animals and cultured tumor cells in order to make correct conclusions.

2) All the results are consistent with surface detection of PRL-3 in cells dying as a consequence of nutrient deprivation or rapid cell division such those that encode metastatic potential. Indeed in the B16 melanoma experiment (Supplementary Figure 2) it is the rapidly dividing cell population that becomes positive for surface PRL-3. It should be determined how expression of PRL-3 on cell surface vs. cytoplasm correlates with Ki67 and cell division (by CFSE or MTV staining).

3) In starved cells, which also upregulate expression of "surface" PRL-3, this finding should be assessed in conjunctions with molecular features of survival vs. autophagy and apoptosis.

4) Figure 5: The investigators determined that PRL-3-zumab induced recruitment of NK cells, B cells and macrophages in the tumor microenvironment mediated via Fc receptor binding. For identification of M1 vs. M2 macrophages only expression of CD86 and CD206 was assessed. This is very superficial assessment of M1 vs. M2 identity. A complete signature panel of M1 and M2 markers should be assessed in order to support this claim substantially.

5) Figure 6: Expression of PRL-3 in various primary patients' tumors was assessed by immunoblot. Based on the studies shown in mouse tumors, expression of PRL-3 on the cell surface does not necessarily correlate with intracellular expression of PRL-3. In order to support the conclusion that PRL-3 expressed in human cancers is a candidate target for immunotherapy, the investigators should examine surface expression of PRL-3 in these primary tumor cells.

Point-to-point responses to reviewers' comments

Reviewer 1:

*The manuscript is well written and the experiments are described in great detail. The results support the conclusions drawn. The statistical treatments of the results are appropriate. The only criticism is that the manuscript is very long. For example the Introduction could be cut back to 3 pages. One of the most important elements of this manuscript is that *in vitro* assessments are not always “truthful”. The relationship between microenvironment stresses leading to surface expression is important in its own right.*

We would like to express our heartfelt gratitude to this highly supportive reviewer for his/her kind efforts and precious time in reviewing this manuscript. We really appreciate his/her compliments to encourage and support our unconventional novel cancer therapy, are most delighted that the reviewer agrees on the shortcomings of *in vitro* assessments as not always being “truthful” of the *in vivo* environment, particularly with regards to surface expression of PRL-3. As advised, we have significantly shortened our manuscript in several parts, particularly the Introduction (reduced to < 2 pages) and Discussion (reduced to < 3 pages) sections. Figure legends have also been shortened where possible.

Reviewer 2:

The authors found that in tumor cells isolated from tumor bearing mice but not in counterparts of the same tumor cells cultured in vitro, PRL-3 can be detected on the cell surface. They support that the extracellular localization of PRL-3 in the tumor cells is a consequence of exosome formation and can be a therapeutic target for cancer treatment because an antibody against PRL-3 recruited NK cells, B cells and M1 macrophages in the tumor microenvironment by Fc receptor-mediated mechanism and decreased tumor growth. The results are interesting and have therapeutic potential.

We sincerely thank the reviewer for his/her appreciation of our study and its therapeutic potential, which present a mechanistic basis for the efficacy and specificity of immunotherapy against 'intracellular' PRL-3 as a new frontier for cancer therapy.

Specific points:

1. *The assessment of PI negative cells in Figure 2c is inaccurate because the numbers and histograms shown in the upper left quadrants of the right and left panels in 2c are incompatible.*

We sincerely apologize for the lack of clarity in our presentation of these results. The original numbers actually represented the mean percentage \pm SD of live, surface+ cells (top left quadrant) out of the total live surface+ and live surface- cell population (top left and bottom left quadrants, respectively), as derived from four biological replicates. In our original presentation, we did not include dead cells (right quadrants in all panels) in our calculations. While the calculations are correct, we have improved our presentation now by relabeling each dot plots to clearly demarcate 'Live' vs 'Dead' populations and 'Surface+' vs 'Surface-' populations (**updated Main Figures 2b-d**). The representative percentage of cells is now shown for each quadrant. Finally, a new table presents the calculated surface+ (% of total live cells) below the dot plots (**updated Main Figure 2e**). We hope the reviewer will find the new Figure presentation and the more detailed description in the text and Figure Legend much clearer now.

Updated Main Figures 2b-e. Key changes are highlighted in blue. PI- and PI+ cells have been relabeled as 'Live' and 'Dead', respectively.

Formula used: % Surface+ = Live Surface+ / total live cells (Surface+ and Surface-)

Note: Mean ± SD values presented in **Main Fig. 2e** for each antigen were from 4 independent experiments, each calculated as described above.

- The tumor cells included in the left upper quadrant of the right hand side panels (labelled 84.4% in c, and 64.4% in d) is most likely a population of proapoptotic tumor cells on their way to undergo cell death and become PI positive. The authors should assess both annexin V and PI and exclude apoptotic and dead cells.

We thank the reviewer for this insightful suggestion. As our analysis of live surface+ tumor cells (upper left quadrants of 'Tumor cells' in Main Figs. 2c and 2d) might indeed include both viable and early apoptotic cells, we have repeated our analysis using Annexin-V to distinguish whether the surface+ population were viable (Annexin-V negative) or early apoptotic (Annexin-V positive). Our data shows that

around 15-25% of live surface+ cells for both positive control EGFR (**new Supplementary Figure 3b**) and PRL-3 (**new Supplementary Figure 3c**) were Annexin-V negative (viable), with the remainder staining positive for Annexin-V (early apoptotic). The relatively high percentage of early apoptotic cells seen for both antigens may be attributed to mechanical stress during the processing of the excised liver tumor tissues to yield single cells for flow cytometry which, for liver tissues, required subjecting them to an extended “tough” tissue dissociation protocol as recommended by the manufacturer (Miltenyi Biotec). We propose that the presence of ‘surface PRL-3’ on viable tumor cells may serve as a key trigger for PRL3-zumab binding and activation of classical antibody-mediated immune clearance pathways in vivo, including ADCC/ADCP, which can further enhance PRL-3 externalization and tumor targeting (feedforward “kill-and-leak” ADCC/ADCP cascade; **Main Figure 5d**).

New Supplementary Figures 3b and 3c. Breakdown of surface+ EGFR live tumor cells (**b**) and surface+ PRL-3 live tumor cells (**c**) into viable (Annexin-V negative) and early apoptotic (Annexin-V positive) populations.

- Rapidly proliferating tumor cells undergo cell death and this is likely the reason for which become positive for “surface” expression of PRL-3. The authors should assess proapoptotic, apoptotic and dead cells in tumors isolated from animals and cultured tumor cells in order to make correct conclusions.*

Many thanks for this insight. To study the possible relationship between proliferation rate, cell viability, and surface PRL-3 expression, we analyzed surface PRL-3 expression on several cell lines known to have varying proliferation rate and PRL-3

expression status: B16F0 murine melanoma cells (PRL-3 positive), HCT116 human colorectal carcinoma cells (PRL-3 positive), MHCC human liver cancer cells (PRL-3 positive), and MKN45 human gastric cancer cells (PRL-3 negative) (**Figure R1a and R1b**). Proliferation rate was measured using an MTS proliferation assay (Promega), viability was determined using Live/Dead staining, and total cellular PRL-3 expression was quantitated using immunoblot band densitometry using ImageJ. Our results show a significantly positive correlation between surface PRL-3 expression and proliferation rate in these cells ($r^2 = 0.997$, $P = 0.003$). In contrast, a borderline correlation was observed between surface PRL-3 and cellular viability ($r^2 = 0.927$, $P = 0.073$), and no significant correlation was found between surface PRL-3 and total cellular PRL-3 expression ($r^2 = 0.705$, $P = 0.295$). Taken together, our results support the reviewer's hypothesis that more rapidly proliferating cells have higher tendency of surface PRL-3 expression. Notably, we did not observe a relationship between rapid proliferation, elevated cell death, and enhanced surface PRL-3 expression within this group, as the top 2 proliferating cell lines with highest surface+ PRL-3 expression (B16F0, HCT116) had the *lowest* cell death (>93% viable; **Figure R1**).

Figure R1. (a) Immunoblot of PRL-3 expression in various cancer cell lines. (b) Summary of relative cellular PRL-3 expression (“PRL-3 WB”), relative proliferation rate, viability, and surface PRL-3 expression of cells in (a).

As advised by the reviewer, we next analyzed the expression of surface PRL-3 on proapoptotic, apoptotic and dead cells in B16F0 cultured cells and tumor cells isolated from animals using Annexin-V and a Live/Dead stain. IgG served as a non-specific control. Surface PRL-3 was specifically detected on both viable and early apoptotic cultured cells, with higher percentage of early apoptotic cells expressing surface PRL-3 (9.88% in viable cells as compared to 41.0% in early apoptotic cells;

Figure R2a). A similar pattern of surface+ PRL-3 was observed in tumor cells (52.8% in viable cells as compared to 87.7% in early apoptotic cells; **Figure R2b).** We constantly observed higher PRL-3 surface+ cell populations in tumor cells than in cultured cells, which partially explains PRL3-zumab’s efficacy in *in vivo* tumor microenvironments to specifically block PRL-3 positive (but not negative) tumors. Both dead cells and early apoptotic cells demonstrated significant nonspecific staining with IgG negative control antibodies, with greatest nonspecific staining in dead cells due to total loss of plasma membrane integrity (**Figure R2).** Taken together, our results demonstrate that *PRL-3 is specifically expressed on viable PRL-3-positive cultured and tumor cells*, and supports our proposed model of elevated PRL-3 on viable tumor cells as the primary trigger or “fire starter” for PRL3-zumab binding to activate immune system’s classical antibody-mediated immune pathways for tumor clearance (**Main Figure 5d).**

Figure R2. Surface PRL-3 is specifically expressed on viable cells. (a,b) Surface expression profiles of PRL-3 and EGFR in viable, early apoptotic, and dead B16F0 cultured cells (a) and tumor cells (b). Viable cells (red boxes) possessed highly specific staining of surface PRL-3 relative to the nonspecific (IgG) staining, as compared to early apoptotic and dead cells with moderate and high nonspecific staining, respectively.

4. All the results are consistent with surface detection of PRL-3 in cells dying as a consequence of nutrient deprivation or rapid cell division such those that encode metastatic potential. Indeed in the B16 melanoma experiment (Supplementary Figure

2) *it is the rapidly dividing cell population that becomes positive for surface PRL-3. It should be determined how expression of PRL-3 on cell surface vs. cytoplasm correlates with Ki67 and cell division (by CFSE or MTV staining).*

We thank the reviewer for the suggestion. In our response to Comment 3 above, we indeed demonstrated a positive correlation between the proportion of PRL-3 surface+ cells and proliferation rate ($r^2 = 0.997$, $P = 0.003$). However, no correlation was observed between surface PRL-3 expression and total PRL-3 expression ($r^2 = 0.705$, $P = 0.295$). To study this further, we heeded the reviewer’s suggestion and studied the relationship between surface PRL-3 and Ki-67, an intracellular protein whose expression is tightly associated with cell proliferation¹. We found that 99.3% of cultured cells and 89.4% of tumor cells with surface PRL-3 expressed Ki-67+ (**Table R1**). These results indicate that PRL-3 is preferably externalized when in an actively proliferating state (Ki-67+). However, active proliferation (i.e. Ki-67+ expression) alone was insufficient for surface PRL-3 externalization, as there were globally less surface PRL-3+ cells than Ki-67+ cells both in cultured cells (7.22% vs 87.4%, respectively) and tumor cells (31.5% vs 86.7%, respectively; **Table R1**). We thus conclude that surface PRL-3 expression is regulated by additional factors than proliferation alone and seek the reviewer’s understanding that it is beyond the scope of the current manuscript to address this in greater detail.

Table R1. Correlation of surface PRL-3 expression and Ki-67 staining in B16F0 cultured cells and tumor cells. Results are background corrected and presented as mean \pm SD ($n = 3$).

	Cultured cells	Tumors
% Surface PRL-3 ⁺	7.22 \pm 2.92	31.5 \pm 1.90
% Ki-67 ⁺	87.4 \pm 9.17	86.7 \pm 3.11
Surface PRL-3, Ki-67 correlation		
PRL-3 ⁺ Ki-67 ⁺	99.3 \pm 0.54	97.1 \pm 0.39
PRL-3 ⁺ Ki-67 ⁻	0.68 \pm 0.54	2.93 \pm 0.39

5. *In starved cells, which also upregulate expression of “surface” PRL-3, this finding should be assessed in conjunctions with molecular features of survival vs. autophagy and apoptosis.*

Thanks for this insight. Herein, we assessed the expression of PRL-3 and several molecular hallmarks of survival, apoptosis and autophagy in both normal and serum starved MHCC-LM3 cells (**new Supplementary Figure 4**). Our results show that serum starvation induces the activation of apoptosis and autophagy, concomitant with an increase in the proportion of PRL-3- surface+ cells.

New Supplementary Figure 4. Starvation induces PRL-3 accumulation and concurrently activates survival, apoptosis and autophagy pathways in MHCC-LM3 cells. Molecular markers of each pathway were tested by immunoblotting. GAPDH, loading control.

Survival markers. We observed upregulated expression levels of the canonical Bcl-2 family pro-survival markers Bcl-2 and Bcl-xL (**new Supplementary Figure 4**). It was previously shown that Bcl-2 plays an essential role to maintain the viability of cells and reduce apoptotic cell death under serum deprivation conditions². We hypothesize that the upregulation of Bcl-xL serves a similar purpose in this case, i.e. a

mechanism by cancer cells to counter death and prolong survival under such serum-starved conditions.

Apoptosis markers. As expected, we observed a pronounced increase in PARP cleavage in starved cells, corresponding with a concomitant decrease in full-length PARP (**new Supplementary Figure 4**). Likewise, there was a prominent accumulation of cleaved caspase-3 (**new Supplementary Figure 4**). As both PARP and caspase-3 are actively cleaved during active apoptosis, these observations indicate that serum starvation induced active apoptosis in these cells. Likewise, cellular levels of Bax, a proapoptotic Bcl-2 family member, were upregulated in starved cells (**new Supplementary Figure 4**).

Autophagy markers. We observed an accumulation of LC3B-I and LC3B-II in starved cells, concomitant with a downregulation of the autophagy substrate SQSTM1/p62 (**new Supplementary Figure 4**), demonstrating enhanced autophagy in starved cells relative to cells cultured under normal conditions.

As the increase (8.4-fold) in the PRL-3 surface+ cell population upon serum-starvation (**Main Fig. 2g**) is greater than the upregulation of total cellular PRL-3 levels in starved cells (2-fold; **new Supplementary Figure 4**), we reason that the increase PRL-3 surface+ cell population is primarily attributable to enhanced PRL-3 *externalization* rather than increased PRL-3 *expression* under starvation conditions, and may be associated with either survival, apoptosis, and/or autophagy pathway activation.

6. *Figure 5: The investigators determined that PRL-3-zumab induced recruitment of NK cells, B cells and macrophages in the tumor microenvironment mediated via Fc receptor binding. For identification of M1 vs. M2 macrophages only expression of CD86 and CD206 was assessed. This is very superficial assessment of M1 vs. M2 identity. A complete signature panel of M1 and M2 markers should be assessed in order to support this claim substantially.*

We thank the reviewer for this advice. We agree that using CD86 and CD206 alone might be a superficial assessment to conclude M1 vs M2 macrophage identity, as these surface antigens are not restricted to M1 or M2 macrophages alone and can be expressed on other immune cell types. Herein, we performed immunofluorescence analysis of tumor sections using a new antibody against the well-accepted pan-macrophage marker, F4/80³. Our results showed that F4/80⁺ cell infiltration into tumors was significantly elevated in the PRL3-zumab treated group but not in other groups ($P < 0.0010$, one-way ANOVA; **updated Main Figs. 5c and 5d**).

Updated Main Figure 5. Changes are highlighted within the blue boxed regions.

To validate this, we obtained fresh tumor isolates from each treatment group and analyzed these using a myeloid immunoprofiling panel to study macrophage infiltration. The gating procedure excluded lineage-specific CD3⁺ (T cells), CD45R/B220⁺ (B cells) and CD335⁺ (NK cells) from the total CD45⁺ population, thus ameliorating the specific signals of lineage-negative (CD45⁺Lin⁻) cell populations. In this analysis, a significantly higher accumulation of tumor-infiltrating CD11b⁺Lys-6C⁺Lys-6G⁻ monocytes/macrophages was reproducibly observed in PRL3-zumab treated tumors, but not in the other groups (**new Supplementary Figures 7a and 7b**). As previous studies have demonstrated that CD11b⁺Lys-6C⁺ tumor-associated monocytes/macrophages specifically express F4/80 antigen⁴, these immunoprofiling results are consistent with our immunofluorescence results in **Main Figure 5c**. In the revised manuscript, we have softened our conclusion and highlighted only the FcR-dependent accumulation of macrophages in PRL3-zumab treated tumors (**updated Main Figs. 5c and 5d**), without specifying the subtypes involved. Understanding from the reviewer is much appreciated.

New Supplementary Figure 7a and 7b. PRL3-zumab promotes the accumulation of CD11b⁺Lys-6C⁺Lys-6G⁻ monocytes/macrophages in liver tumors in an FcR-dependent manner. (a) Immunoprofiling of live infiltrating CD45⁺Lin⁻CD11b⁺Lys-6C⁺Lys-6G⁻ monocytes/macrophages in orthotopic MHCC-LM3 liver tumor extracts from mice subjected to various treatments. Monocyte/macrophage gates are highlighted with red boxes. (b) Summary of the mean percentage of live CD45⁺Lin⁻CD11b⁺Lys-6C⁺Lys-6G⁻ monocytes/macrophages for each treatment group in (a).

7. *Figure 6: Expression of PRL-3 in various primary patients' tumors was assessed by immunoblot. Based on the studies shown in mouse tumors, expression of PRL-3 on the cell surface does not necessarily correlate with intracellular expression of PRL-3. In order to support the conclusion that PRL-3 expressed in human cancers is a candidate target for immunotherapy, the investigators should examine surface expression of PRL-3 in these primary tumor cells.*

We appreciate this excellent point by the reviewer. The timeframe required for regulatory approval from National Cancer Center Singapore for the suggested study of primary tumor cells posed us a challenge to obtain fresh human tumor samples within the revision deadline. Nonetheless, we were fortunate to obtain a patient-matched pair of freshly-excised kidney tumor sample and nontumor kidney tissue from one of our clinical collaborators in time for the resubmission. Pathological evaluation confirmed the absence of tumor cells in the nontumor tissue (hRCC1N), whilst >90% of the tumor sample demonstrated salient features of clear cell renal carcinoma, without any normal kidney parenchyma notable (hRCC1T; **new Supplementary Figure 8a**). In line with PRL-3 protein as a tumor-specific oncotarget, PRL-3 was absent in the nontumor kidney tissue (hRCC1N), but strongly expressed in the freshly-excised kidney tumor tissue (hRCC1T; **new Supplementary Fig. 8b**). Importantly, surface PRL-3 levels (as determined by flow cytometry) followed a similar pattern, with surface PRL-3 detected on dissociated tumor cells, but not on nontumor cells (**new Supplementary Figs. 8c and 8d**).

Supplementary Figure 8. PRL-3 is expressed on primary human kidney tumor cells.

(a) Hematoxylin and eosin stain of patient-matched nontumor kidney samples (hRCC1N; upper panel) and primary clear cell renal carcinoma samples (hRCC1T; lower panel). Bar, 400 μ m. (b) WB for PRL-3 expression. GAPDH served as a loading control. (c) Surface PRL-3 expression profiles using PRL3-zumab. (d) Background corrected values of PRL-3 surface+ cell populations after subtracting nonspecific IgG binding for each sample.

References

- Scholzen, T. & Gerdes, J. The Ki-67 protein: from the known and the unknown. *J Cell Physiol.* **182**, 311-322 (2000).
- Xu, H.D. et al The pro-survival role of autophagy depends on Bcl-2 under nutrition stress conditions. *PLoS one.* **8**, e63232 (2013).
- Gordon, S., Hamann, J., Lin, H.H. & Stacey, M. F4/80 and the related adhesion-GPCRs. *Eur J Immunol.* **41**, 2472-2476 (2011).
- Movahedi, K. et al Different Tumor Microenvironments Contain Functionally Distinct Subsets of Macrophages Derived from Ly6C(high) Monocytes. *Cancer Res.* **70**, 5728-5739 (2010).

Reviewers' comments:

Reviewer #1 (Remarks to the Author):
(No additional comment for the authors)

Reviewer #2 (Remarks to the Author):

In the revised manuscript Thura et al. have addressed most of my previous concerns and have revised their manuscript appropriately. However serious issues still remain unresolved and require further experimentation and attention.

Specific issues:

- 1) Lines 329-337, Figure 5c and Supplementary Figures 7a and 7b: Figure 5c shows that significantly higher F4/80+ macrophages were recruited in PRL3-zumab treated tumors (Group II). In Supplementary figure 7, the authors show that the PRL3-zumab treated tumors (Group II) also have higher recruitment of CD45+Lin-CD11b+Lys-6C+Lys-6G- and assume that this population represents macrophages. This conclusion is not supported by experimental data. In fact, it is well established that CD45+Lin-CD11b+Lys-6C+Lys-6G- are M-MDSC, which represent one of the major populations of myeloid suppressor cells. In order to conclude that these cells are not M-MDSC but macrophages, the authors should assess whether this myeloid cell population has suppressor function and whether CD45+Lin-CD11b+Lys-6C+Lys-6G- cells express F4/80 or not.
- 2) Supplementary Figure 8d: It is unclear how the authors came to the numerical values of PRL-3 surface expression in this single tumor sample that they have assessed. The flow cytometry results shown in panel c show a 2-fold difference in live cells with surface expression of PRL-3 between normal kidney tissue and kidney cancer. In the figure legend it is indicated that the numbers of panel d are based on calculated values after subtracting nonspecific IgG binding for each sample. These experimental data of IgG binding should be included in the figure. Without these, the reported values are not compelling because they are contradictory to the shown experimental results.
- 3) Figure 6: The results shown in Figure 6 were all generated by western blots and do not support the authors' claims that PRL-3 might be a candidate target for immunotherapy. Surface expression in tumor cells representing these cancer types should be documented. Otherwise, the conclusions supported by the authors are not consistent with the provided experimental evidence.

Point-to-point responses to Reviewers' comments

Reviewer 1:

No additional comment for the authors.

Many thanks to this reviewer for his/her full endorsement of our revised manuscript.

Reviewer 2:

In the revised manuscript Thura et al. have addressed most of my previous concerns and have revised their manuscript appropriately.

We are thankful to the Reviewer for his/her overall satisfaction with our revised manuscript. We are delighted that through our hard efforts, the Reviewer is happy with our manuscript revisions which addressed most of his/her previous concerns. In this new revision, we further carefully address the additional 3 new points raised:

Specific issues:

1. ***Lines 329-337, Figure 5c and Supplementary Figures 7a and 7b:*** *Figure 5c shows that significantly higher F4/80+ macrophages were recruited in PRL3-zumab treated tumors (Group II). In Supplementary figure 7, the authors show that the PRL3-zumab treated tumors (Group II) also have higher recruitment of CD45+Lin-CD11b+Lys-6C+Lys-6G- and assume that this population represents macrophages. This conclusion is not supported by experimental data. In fact, it is well established that CD45+Lin-CD11b+Lys-6C+Lys-6G- are M-MDSC, which represent one of the major populations of myeloid suppressor cells. In order to conclude that these cells are not M-MDSC but macrophages, the authors should assess whether this myeloid cell population has suppressor function and whether CD45+Lin-CD11b+Lys-6C+Lys-6G- cells express F4/80 or not.*

Many thanks to the Reviewer for this insight. Please see **updated Supplementary Fig. 7** below, which directly answers the second half of the Reviewer's comment. Essentially, to validate the macrophage identity of the CD45⁺Lin⁻CD11b⁺Ly-6G⁻Ly-

6C⁺ population, we reassessed the immunoprofiles of tumor infiltrating cells by including an anti-F4/80 antibody (clone BM8) in our panel. We have updated our Results to highlight the accumulation of Ly-6C⁺F4/80⁺ macrophages in the **Main Text (lines 327-339)**. The flowchart below summarizes the sequential categorization of various myeloid cell populations¹, with the CD11b⁺Ly-6G⁻Ly-6C⁺ myeloid population in question highlighted within the boxed region:

Myeloid cells immunoprofiling flowchart

Updated Supplementary Figures 7a and 7b. (a) Increased infiltration of Ly-6C⁺F4/80⁺ macrophages (red boxed regions) specifically in PRL3-zumab treated tumors. Immunoprofiling of live infiltrating CD45⁺Lin⁻CD11b⁺Ly-6G⁻ cells in orthotopic MHCC-LM3 liver tumor extracts from mice subjected to various treatments. (b) Summary of the mean percentage ± SEM of live Ly-6C⁺F4/80⁺ macrophages for each group in (a). P-value was calculated using one-way ANOVA.

The Reviewer is right in that there was also an enrichment of Ly-6C⁺F4/80^{low} myeloid cells (including monocytes and/or M-MDSCs) in PRL3-ZUMAB-treated samples (**Fig. R1**):

Figure R1. Summary of the mean percentage \pm SEM of live Ly-6C⁺F4/80^{low} myeloid cells for each treatment group in updated Supplementary Fig. 7a. P-value was calculated using one-way ANOVA.

The Ly-6C⁺F4/80^{low} population comprises both monocytes and M-MDSCs, but it is challenging to segregate between the two using flow markers due to similarities in their phenotype¹. M-MDSCs can only reliably be defined by their functional suppressive activity, which is absent in normal monocytes². Classical M-MDSC functional assays based on suppression of autologous or allogenic T cells are not applicable to our orthotopic liver tumor models in the athymic *Ncr nude* mouse strain, as these mice are T cell deficient. Thus, functional assays on alternative target cell populations must to be developed to answer this – an endeavor that necessitates expertise beyond our team’s technical familiarity. Even if we keep trying for a prolonged period to establish such functional assays, the data generated are still ultimately *in vitro* findings, which may not accurately recapitulate the *in vivo* situation.

For the past decade, we have been focusing on clinically-relevant *in vivo* tumor models for novel cancer therapy. Herein, we seek the Reviewer’s consideration of the positive therapeutic endpoints of PRL3-zumab immunotherapy. M-MDSCs

are regarded as a *pro-tumorigenic* myeloid cell population playing roles in promoting tumor growth, angiogenesis, and metastasis³. As we showed that PRL3-zumab therapy is Fc-dependent, we logically believe PRL3-zumab recruits activator (rather than suppressor) myeloid cell populations into tumor areas to block (rather than promote) tumor growth. Regardless, we will certainly keep the Reviewer's suggestions in our future experiments as we address this in collaboration with Immunology researchers, but such detailed analysis will be beyond the scope of the current manuscript. The Reviewer's understanding is sincerely appreciated to facilitate the timely publication of this important study. Nonetheless, if the Reviewer is not satisfied with the flow data presented, we are willing to remove Supplementary Figures 7a and 7b altogether, as they do not affect the critical conclusions of our study.

2. **Supplementary Figure 8d:** *It is unclear how the authors came to the numerical values of PRL-3 surface expression in this single tumor sample that they have assessed. The flow cytometry results shown in panel c show a 2-fold difference in live cells with surface expression of PRL-3 between normal kidney tissue and kidney cancer. In the figure legend it is indicated that the numbers of panel d are based on calculated values after subtracting nonspecific IgG binding for each sample. These experimental data of IgG binding should be included in the figure. Without these, the reported values are not compelling because they are contradictory to the shown experimental results.*

We apologize for this oversight. We have herein included the experimental data of IgG binding in **updated Supplementary Fig. 8c**. The revised table (**Supplementary Fig. 8d**) summarizes the calculated surface+ (% of total live cells) for each dot plot, as well as the background corrected ("Adj. PRL-3" = %Surface PRL-3 - %Surface IgG) values for both normal and tumor kidney samples. We hope the reviewer will find the new Figure presentation and the basis of the calculated numerical values of PRL-3 surface expression clearer now.

d

Sample	PRL-3 WB	Surface+ (% of live cells)		
		IgG	PRL-3	Adj. PRL-3
Normal	Negative	3.48	3.07	0.00 [#]
Tumor	Positive	3.77	6.26	2.48

[#]Values below control (hIgG) background staining reflected as zero

Updated Supplementary Figures 8c and 8d. PRL-3 is detected on the surface of freshly-operated human kidney tumor cells. (c) Surface PRL-3 expression profiles using control hIgG (upper panels) or PRL3-zumab (lower panels). (d) Mean percentage surface positive (surface+) live cells for each panel in (c) were calculated by dividing the surface antigen-positive live cells (upper left quadrant) by total live cells (sum of both upper and lower left quadrants). The background adjusted values of surface PRL-3 (“Adj. PRL-3”) were calculated using the formula $\text{Adj. PRL-3} = (\% \text{surface+ PRL-3}) - (\% \text{surface+ IgG})$ and presented in the rightmost column. For normal kidney cells, surface PRL-3-specific staining was lower than negative control IgG staining, and thus reported as zero.

3. **Figure 6:** *The results shown in Figure 6 were all generated by western blots and are do not support the claims authors’ claims that PRL-3 might be a candidate target for immunotherapy. Surface expression in tumor cells representing these cancer types should be documented. Otherwise, the conclusions supported by the authors are not consistent with the provided experimental evidence.*

We appreciate the reviewer’s suggestion. However, due to the scarcity of freshly-excised human tumors available to us from the limited pool of cancer patients in local hospitals for such analysis, we face uncertainty in the time taken for establishing multiple new collaborations with different hospital surgeons as well as securing ethics approval to complete such a representative study for surface PRL-3 expression across the 11 tumor types. This may delay the paper indefinitely,

and affect the timeliness of our findings. In view of this, we have instead updated our Results and Discussion sections as follows:

Lines 348-349: Shortened the subheading to “PRL-3 is overexpressed in diverse human cancers”

Lines 382-385: Deleted the sentences: “In summary, our results propose PRL-3 as an excellent, accessible oncotarget in multiple cancer types, particularly in aggressive tumors with urgent, unmet medical needs. PRL3-zumab will serve as a candidate for novel cancer therapy against such PRL-3-positive tumors.”

Lines 446-448: Shortened the final paragraph to “PRL-3 is specifically overexpressed in 80.6% randomly-analyzed fresh-frozen human tumors (across 11 cancer types), but not in any matched normal tissues examined. Moreover, PRL3-zumab appears well-tolerated by cancer patients in an ongoing Phase 1 trial in Singapore (<https://clinicaltrials.gov/ct2/show/NCT03191682?term=PRL-3&rank=1>). Such a safety profile, coupled with the preclinical efficacy of PRL3-zumab in orthotopic tumor models, lends support for further studies of PRL3-zumab in PRL-3-positive cancer patients as a safe, promising treatment modality.”

In view of the experiments showing that PRL-3 is externalized on the surface in both liver and metastatic melanoma mouse models, as well as on freshly isolated primary human kidney tumor cells and primary human nasal tumor cells (see **Additional Information** below), it is likely that PRL-3-zumab may be explored for possible treatment in multiple human cancers, although the externalization of PRL-3 in the various human cancers need to be validated in future experiments. We sincerely appreciate the understanding from this Reviewer.

Additional Information

Although Head & Neck tumors were not in our list of tumor types shown in Fig. 6, we were fortunate to be granted a tiny piece of freshly-excised nasal tumor sample offered by an existing collaborator. Our analysis demonstrated the presence of surface PRL-3 on individual nasal tumor cells (**Fig. R2**). Unfortunately, due to the tiny tissue size, we used the entire sample to harvest sufficient cells for flow analysis and could not obtain tissue lysate to correlate PRL-3 expression levels by Western blotting.

Figure R2. PRL-3 is expressed on the surface of freshly-operated nasal tumor cells. Surface PRL-3 expression profiles using control hlgG (upper left panel) or PRL3-zumab (upper right panels). Mean percentage surface positive (surface+) live cells were calculated by dividing the surface antigen-positive live cells by total live cells. The background adjusted values of surface PRL-3 (“Adj. PRL-3”) were calculated using the formula $\text{Adj. PRL-3} = (\% \text{surface+ PRL-3}) - (\% \text{surface+ IgG})$ and presented in the rightmost column.

Importantly, our past decade of unconventional cancer therapy studies using PRL-3 antibodies to inhibit tumors expressing intracellular PRL-3 oncotarget in multiple animal models using various cancer cell lines deriving from different tissue origins, such as ovarian, colorectal, melanoma, lung, AML, gastric, and liver firmly support the scientific principle that, regardless of tumor type, PRL-3 antibody specifically blocks PRL-3-positive (but not PRL-3-negative) tumors *in vivo*. (**Table R1**). Taken together with the demonstrated requirement for Fc interactions for therapeutic outcome, these observations suggest that PRL-3 is expressed on the surface of these multiple tumor types as a target for immunotherapy *in vivo*.

Table R1: List of publications between 2008 to date demonstrating that PRL-3 antibody therapy can unconventionally block PRL-3-positive tumors, but not PRL-3-negative tumors which do not express PRL-3 target

Cell line used for tumor model	Cell line origin	PRL-3 expression	Anti-tumor response with PRL-3 antibody therapy?	Reference(s)
CHO-PRL-3	Ovarian	Positive	Yes	Cancer Biol. Ther. 2008, 7(5):750-7
CT26	Colorectal	Negative	No	
B16F0	Melanoma	Positive	Yes	Sci. Trans. Med. 2011, 3(99): 99ra85
B16F10	Melanoma	Negative	No	
HCT116	Colorectal	Positive	Yes	Oncotarget 2012, 3(2): 158–171
A2780	Ovarian	Positive		
NCI-H460	Lung	Negative	No	
TF1-ITD	AML	Positive	Yes	EMBO Mol. Med. 2013 5(9): 1351-66
SNU-484	Gastric	Positive	Yes	JCI Insight , 2016, 1(9): e87607
IM-95	Gastric	Positive		
NUGC-4	Gastric	Positive		
MKN45-PRL-3	Gastric	Positive		
MKN45	Gastric	Negative		
MHCC-LM3	Liver	Positive	Yes	Current manuscript (in revision)
Hep53.4-PRL-3	Liver	Positive		
Hep53.4	Liver	Negative	No	

In the current manuscript, we demonstrate that surface+ PRL-3 live cell populations could be detected on PRL-3-positive kidney tumor cells, but not PRL-3-negative normal kidney cells (**Supplementary Fig. 8d**). Although not one of the 11 tumor types shown in Main Fig. 6, the additional nasal tumor sample (**Fig. R2**) also demonstrated surface PRL-3 expression. In this revised manuscript, we further included new results from 8 additional pairs of gastric tumor-gastric normal samples, with 7 out of the 8 gastric tumors shown to be PRL-3 positive. Nonetheless, taking into account the reviewer’s suggestion that PRL-3 protein expression data by Western blot may not fully support PRL-3 as a surface target for immunotherapy, we have softened our conclusion to report the high frequency of PRL-3 expression in multiple tumors (80.6%, after including 8 new gastric tumor samples).

References

1. Richards, D.M., Hettinger, J. & Feuerer, M. Monocytes and macrophages in cancer: development and functions. *Cancer Microenviron.* **6**, 179-191 (2013).
2. Bronte, V. et al Recommendations for myeloid-derived suppressor cell nomenclature and characterization standards. *Nat Commun.* **7**, 12150-12150 (2016).
3. Marvel, D. & Gabrilovich, D. Myeloid-derived suppressor cells in the tumor microenvironment: expect the unexpected. *J Clin Invest.* **125**, 3356-3364 (2015).

REVIEWERS' COMMENTS:

Reviewer #2 (Remarks to the Author):

The authors have addressed my previous comments, performed additional work and revised the manuscript accordingly.

Response to Reviewers

Reviewer #2: The authors have addressed my previous comments, performed additional work and revised the manuscript accordingly.

We sincerely thank this Reviewer for his full support on our revised manuscript for publication in Nature Communications.